# Study on the allocation efficiency of medical and health resources in Hainan Province: Based on the super-efficiency SBM—Malmquist model

**Yanhua Gong**[‡]*, **Dong Ma**[‡], **Wen Feng**

School of Management, Hainan Medical University, Haikou City, Hainan Province, China

‡ YG and DM are contributed equally to this work as co-first authors.
* 1532264196@qq.com

**Data Availability Statement:** The basic data of this paper can be obtained from Hainan Statistical Yearbook (2017-2021) and National Economic and Social Development Statistical bulletins of Hainan

## Abstract

The equity and efficiency of medical and health resource allocation is the key point of health reform in all countries. Poor allocation efficiency of health resources will seriously affect the sustainable and high-quality development of health causes. Hainan Province is the only free trade port with Chinese characteristics in China, which means that Hainan has ushered in a brand-new development under the policy of free trade port. This study aims to adopt policies to improve the efficiency of medical and health resource allocation in Hainan Province and provide references for other regions. In this study, the Super-efficiency SBM and Malmquist models were used to analyze the static and dynamic efficiency of medical and health resource allocation in Hainan Province during 2016–2020. The results showed that, statically, the average efficiency of comprehensive allocation of health resources in Hainan Province from 2016 to 2020 was 0.975, showing poor overall performance and significant regional differences. Dynamically, the average index of allocation efficiency of medical and health resources was 0.934, showing a negative growth trend. The technical efficiency and scale efficiency of health resource allocation efficiency showed positive growth, while the technical progress and pure technical efficiency showed negative growth. It shows that it is influenced by both scale efficiency and technological progress, among which technological progress is the key factor. Therefore, some policy suggestions are put forward to further optimize the allocation of medical and health resources and improve utilization efficiency.

## Introduction

The contradiction between the limited nature of health resources and people's unlimited needs is prominent [1]. How to make more effective use of limited health resources, meet the needs of people at different levels for health services, optimize the allocation of health resources, and improve the efficiency of their use is a common problem facing global healthcare reform [1, 2]. The 2016 Outline of the "Healthy China 2030" Plan proposes that by 2030, China's health

cities and counties. http://www.hainan.gov.cn/hainan/zfwj/list4.shtml https://wst.hainan.gov.cn/swjw/xxgk/0200/0202/newxxgkzl.html?ClassInfoId=2009.

**Funding:** Our research funds came from the innovative project of graduate students of Hainan Provincial Department of Education (Title: Research on Equity and Efficiency of Healthcare Resource Allocation in Hainan Province under the Construction of Free Trade Port. No. Qhys2022-284). Funded by: Hainan Provincial Department of Education, Hainan Medical University. Recipient: Yanhua Gong, Hainan Medical University. The amount of funding is 3,500 RMB. The funders had no role in study design, data collection and analysis, decision to publish, or preparation of the manuscript.

**Competing interests:** The authors have declared that no competing interests exist.

institution system will be more complete, the development of health undertakings will be more coordinated, healthy lifestyles will be popularized, and the quality of health services and the level of protection will be continuously improved. An important prerequisite for improving health and promoting health equity is the rational allocation and effective utilization of medical and health resources. Improving the efficiency of healthcare resource allocation not only promotes the accessibility and quality of healthcare services but also plays an important role in raising people's living standards and promoting economic growth. Since the implementation of China's new healthcare reform in 2009, the investment in health resources has generally increased and achieved great results.

Research on health resource allocation in China began in the 1980s. With the development of the economy, the rapid increase in population, and the intensification of aging, people have put forward higher requirements for medical and health resources and services, and the government has continuously increased its investment in health resources. The number of health institutions, beds, and health technicians has continued to increase, and the service capacity of health institutions has greatly improved, but the efficiency of the allocation of health resources is still low, which may lead to subsequent increases in the input of health resources and poor output. This not only fails to meet the growing health care needs of the people but also results in a serious waste of health care resources, leading to a rapid increase in health care costs, aggravating the health care burden of the people, restricting the sustained and high-quality development of health care, and running counter to the ultimate goal of the country's health and health development. Ye Yizhong (2022) [3] used the three-stage SBM model and Malmquist index method to statically and dynamically evaluate the allocation efficiency of healthcare resources in 31 provinces (autonomous regions and municipalities directly under the central government, except for Hong Kong, Macao, and Taiwan) in China from 2015 to 2020. His study found that the allocation efficiency of health resources in China in 2015–2020 is low and on a downward trend, with obvious constraints on technological progress, large differences in allocation efficiency between regions and provinces, and significant impacts on the gross domestic product of each region, the number of permanent residents, and the level of urbanization. Xiang Yingru (2020) [4] analyzed the efficiency of health resources at the provincial level in China using the BCC model of DEA and the Malmquist index. The study pointed out that after the implementation of China's health care reform, due to the excessive number of public hospitals and the serious aging of the population, there are still problems such as waste of resources, insufficient output in some areas, and inefficiency. Technological progress is a key factor affecting efficiency. Luo Liangqing (2008) [5] utilized the DEA model to study the production efficiency of healthcare services in China's eastern, central, and western regions and found that China's healthcare service production efficiency is generally low. A large number of research results provide theoretical guidance for the research idea in this paper. From many related studies, it can be concluded that DEA is the most mature and widely used method to study the efficiency of healthcare resource allocation. However, the traditional DEA model has certain limitations in studying healthcare resource allocation efficiency. Therefore, this paper improves the method.

The problem of unbalanced regional development has always existed in China. Hainan Province is currently the only free trade port with Chinese characteristics in mainland China, which is also known as the "test bed of national reform and opening up", which means that the successful experience of Hainan's reforms can be extended to other regions of China, and its construction is of great significance to China [6]. In the context of the free trade port, Hainan will have more development opportunities. At the same time, the efficiency of healthcare resource allocation, as an important indicator for safeguarding people's health, has also received widespread attention. Hainan's free trade port policy plays an important role in the

operation of healthcare, for example, by providing more medical resources, promoting medical technology innovation, improving the quality of medical services, promoting the development of medical tourism, and optimizing the medical management system. Hainan Province (108°37′–111°03′ E, 18°10′–20°10′ N) is located in the southernmost part of China, close to Southeast Asia, with a land area of 35,400 square kilometers and a resident population of 10,123,400 in 2020. In 2018, the Central Committee of the CPC decided to support Hainan to progressively explore and steadily deepen the construction of a free trade port with Chinese characteristics, helping Hainan to deepen its reforms comprehensively. Hainan is committed to building a healthy and long-lived island. Under the free trade port policy, Hainan focuses on system integration and innovation and has made significant breakthroughs in the field of healthcare. with the increase in population, the intensification of aging, and the change of disease spectrum, the current allocation of healthcare resources in Hainan Province is unreasonable, and the quality of healthcare services and the utilization of healthcare resources are inefficient, which seriously restricts the sustainable development of healthcare and wellness. Therefore, it is of great significance for Hainan Province to study how to improve the efficiency of healthcare resource allocation in Hainan Province to meet the people's demand for healthcare services, provide an important guarantee for the high-quality development of Hainan's healthcare industry, and escort the construction of the Free Trade Port. In the context of free trade port, this study adopts the super-efficient SBM model and Malmquist index to study the efficiency of healthcare resource allocation in Hainan Province from 2016 to 2020, analyze the status quo, problems, and reasons, and put forward rationalization suggestions to provide reference for the scientific formulation of healthcare development planning.

## Data and methods

### Source of data

The data comes from the Hainan Statistical Yearbook (2017–2021) and the National Economic and Social Development Statistical bulletins of Hainan cities and counties. Because partial data of Lingao County, Ledong County and Qiongzhong County are missing, these three counties are excluded, and 15 cities and counties including Haikou City are finally selected as research objects.

This paper combines policy background, data availability, and trend changes to choose the time span of 2016–2020 to study the healthcare resource allocation efficiency in Hainan Province. In terms of policy background, since 2016, the Chinese government has proposed a series of policy measures to promote the efficiency of healthcare resource allocation, and the implementation of these policies has had a positive impact on the efficiency of healthcare resource allocation in Hainan Province. In addition, in 2018, under the policy of the Free Trade Port, Hainan Province is fully committed to deepening reform. Therefore, the choice of 2016 as the starting year has a certain policy representativeness, and the choice of 2020 as the cut-off year is because when this study was done, the data could only be obtained up to 2020. In terms of data availability, after 2016, the data related to the allocation of healthcare resources in Hainan Province were gradually improved, which can reflect the actual situation of healthcare resource allocation in Hainan Province in a more comprehensive way, and there were more missing data before 2016. Therefore, we chose the time span of 2016–2020 to ensure the accuracy and reliability of the data as much as possible. In terms of trend change, with the passage of time, the trend of healthcare resource allocation efficiency in Hainan Province will also change, especially in the context of the free trade port. By studying the time span of 2016–2020, the changing trend of healthcare resource allocation efficiency in Hainan Province before and after the implementation of the Free Trade Port can be more clearly grasped, thus providing a reference for policy formulation in Hainan Province.

**Table 1. Description of related indicators.**

| Type | Name | Unit |
|------|------|------|
| **Input indicators** | Institutions [a] | Number |
| | Beds | Sheets |
| | Health Technician | Person |
| **Output indicators** | Visits | Ten thousand person-times |
| | Hospital Admissions | Ten thousand people |
| | Utilization rate of hospital bed | % |
| **Other indicators** | Total health costs | 100 million yuan |
| | Permanent resident population | Ten thousand people |

[a] Institutions mainly include hospitals, primary medical and health institutions, professional public health institutions, internet hospitals, and other medical and health institutions in the Hainan Provincial Health Statistics Yearbook. It reflects the overall size of the region's health systems. It is a public place to provide medical and health services and is one of the carriers.

## Selection of indicators

According to the characteristics of public health input indicators and output indicators and referring to other literatures [7–10], the number of institutions, number of beds and number of health technicians were selected as input indicators, the visits, the hospital admissions and the utilization rate of hospital bed were output indicators, and the total health cost and population were auxiliary indicators (Table 1).

## Super-efficiency SBM

DEA is a non-parametric efficiency evaluation method first proposed by A. Charnes et al. in 1978 [11]. This method does not need to estimate the production function, but only needs to know the corresponding input and output indicators, and can comprehensively evaluate the relative efficiency of units with more input, and more output. Therefore, it has the advantages of a wide application range, easy data acquisition and simple operation. The DEA method includes a variety of different evaluation models. At present, most domestic researchers use the traditional CCR and BCC models to calculate the technical efficiency in the field of health. However, the traditional model to evaluate relative efficiency has the following shortcomings: First, the traditional radial model does not contain the relaxation variable for the measurement of inefficiency, which leads to the error of efficiency measurement. Second, it is impossible to further distinguish the effective decision making unit of DEA [12]. The non-radial model (SBM model) based on relaxation measurement proposed by Tone in 2001 can well solve the relaxation problem [13], but it cannot further distinguish and compare decision units. Then Tone (2002) put forward the super-efficiency SBM model [14], which combines the super efficiency and SBM to solve both the relaxation problem and the decision making unit problem. Jing Gong (2022) and Weilin Liu (2019) et al. also proved that the measurement of super-efficiency SBM is more scientific in their empirical studies on healthcare [7, 15]. Its expression is as follows:

$$\min \rho = \frac{1 + \frac{1}{m}\sum_{i=1}^{m} S_i^- / x_{i0}}{1 + \frac{1}{q}\sum_{r=1}^{q} S_r^+ / y_{r0}}$$

$$St. \sum_{j=1, j \neq k}^{n} x_{ij}\lambda_j + S_i^- \leq x_k$$

$$St. \sum_{j=1, j\neq k}^{n} y_{ij}\lambda_j - S_i^+ \geq y_k$$

$$\lambda_j, S_i^-, S_r^+ \geq 0$$

$$i = 1, \cdots, m; r = 1, \cdots, q; j = 1, \cdots, n(j \neq k)$$

In the above model, $\rho$ is the efficiency value, $n$ is the number of DMU of decision-making unit, $m$ and $q$ respectively are the number of input and output, $S_i^-$ and $S_r^+$ respectively represent the relaxation variables in the model, $\lambda_j$ represents the weight vector of each decision-making unit, $x_{ik}$ and $y_{rk}$ respectively represent the input variables and output variables of each decision-making unit.

According to the super-efficiency SBM model, $\rho < 1$ indicates that the comprehensive efficiency of resource allocation is in a relatively invalid state and the efficiency is low. $\rho \geq 1$ indicates that the comprehensive efficiency of resource allocation is in a relatively effective state and the efficiency is high. It is generally believed that the larger the comprehensive efficiency value, the higher the efficiency of resource allocation, and the better the level.

However, the super-efficiency SBM model also has its limitations. First, this assessment method ignores some non-economic factors, such as environmental efficiency and social efficiency. Therefore, it does not fully assess the comprehensive efficiency. Secondly, it has high requirements for input and output data, but there are also data that are more difficult to obtain in practice, which may affect the assessment results. Third, it cannot reflect the spatial distribution of efficiency.

## Malmquist index

Considering that the super-efficiency SBM model is limited to evaluating the static efficiency of each decision making unit, the degree of efficiency cannot be measured, so this paper introduces the Malmquist exponential model. The Malmquist index model reflects the dynamic change trend of total factor productivity (TFP) by observing the distance between DMU and the production front in different periods. TFP is further decomposed into technical efficiency (EFFCH) and technical progress (TECH), and technical efficiency can be further decomposed into scale efficiency (SECH) and pure technical efficiency (PECH) [7]. Their relationship is as follows:

TFPCH = EFFCH * TECHCH

EFFCH = PECH * SECH.

If the change of each efficiency is greater than 1, it indicates positive growth or improvement. If it's equal to 1, it's not changing. Less than 1 indicates a decrease or decrease.

In this study, data were input and collated by Microsoft Excel, and the super-efficiency SBM and Malmquist index were calculated by MATLAB software.

The Malmquist index assesses technical efficiency change by calculating the product of technical progress and efficiency change. However, the biggest limitation of the method is that it ignores the source of technical progress. In other words, it is not known whether technical progress is driven by internal innovation or external introduction. This problem may lead to errors in the assessment results.

## Results

### Static efficiency

Based on the super-efficiency SBM model, the comprehensive efficiency of health resource allocation in 15 cities and counties in Hainan Province during 2016–2020 was calculated by

**Table 2. Comprehensive efficiency of medical and health resource allocation in 15 cities and counties of Hainan Province from 2016 to 2020.**

| Region | 2016 | 2017 | 2018 | 2019 | 2020 | Mean |
|---|---|---|---|---|---|---|
| Haikou City | 1.248 | 1.264 | 1.223 | 1.112 | 1.134 | 1.196 |
| Sanya City | 0.652 | 1.043 | 1.001 | 0.559 | 0.642 | 0.779 |
| Wuzhishan City | 1.003 | 1.053 | 1.038 | 1.064 | 1.081 | 1.047 |
| Wenchang City | 0.774 | 0.806 | 1.013 | 1.023 | 0.595 | 0.842 |
| Qionghai City | 1.031 | 1.058 | 1.098 | 1.114 | 0.898 | 1.040 |
| Wanning City | 1.087 | 1.033 | 1.016 | 1.026 | 0.590 | 0.950 |
| Ding 'an County | 1.061 | 1.079 | 1.014 | 1.043 | 0.740 | 0.988 |
| Tunchang County | 1.119 | 1.107 | 1.098 | 1.020 | 1.109 | 1.090 |
| Chengmai County | 0.738 | 1.017 | 1.054 | 0.667 | 1.304 | 0.956 |
| Danzhou City | 1.035 | 1.030 | 0.827 | 0.755 | 1.042 | 0.938 |
| Dongfang City | 0.845 | 1.005 | 0.861 | 0.852 | 0.652 | 0.843 |
| Baoting County | 1.156 | 1.098 | 1.138 | 1.137 | 1.170 | 1.140 |
| Lingshui County | 1.115 | 1.069 | 1.164 | 1.115 | 0.657 | 1.024 |
| Baisha County | 0.650 | 0.852 | 0.749 | 1.032 | 0.534 | 0.764 |
| Changjiang County | 0.899 | 1.041 | 1.028 | 1.082 | 1.101 | 1.030 |
| Mean | 0.961 | 1.037 | 1.021 | 0.973 | 0.883 | 0.975 |

using MATLAB software. According to the comprehensive efficiency, it could be seen that the overall level of comprehensive efficiency in all regions was poor from 2016 to 2020. With the average efficiency higher than 1, 7 regions have improved the efficiency of resources for health and medicine, accounting for 46.67%, among which Haikou City has the best efficiency and Baisha County has the lowest efficiency, with obvious regional differences (Table 2).

From 2016 to 2020, the proportion of regions with DEA effective comprehensive efficiency in all sample areas showed a decreasing trend. In terms of each year, there were 9 regions with comprehensive efficiency that reached DEA effective in 2016, accounting for 60%. In 2017, there were 13 regions whose comprehensive efficiency reached DEA effectiveness, accounting for 86.67%. In 2018, the comprehensive efficiency reached DEA effective in 12 regions, accounting for 80%. DEA effective comprehensive efficiency reached 2019 in 11 areas, accounted for 73.33%. In 2020, there were 7 regions with DEA effective comprehensive efficiency, accounting for 46.67%. Among them, Haikou City, Wuzhishan City, Tunchang County and Baoting County have achieved effective comprehensive efficiency for five consecutive years, with rational utilization of input, relatively high efficiency and reasonable allocation of resources. Although there were fluctuations in the comprehensive efficiency of other cities and counties, there would be some years in which the comprehensive efficiency has reached the effective level. There were no cities and counties that have not reached the effective level for five consecutive years (Fig 1).

## Dynamic efficiency

Based on the Malmquist index model, the dynamic changes in health resource allocation efficiency in 15 cities and counties in Hainan Province during 2016–2020 were calculated by using MATLAB software. From 2016 to 2020, the average value of the health resource allocation efficiency index in Hainan Province was 0.934, showing an overall negative growth trend. Specifically, only 2016–2017 witnessed positive growth in total factor productivity, while 2017–2020 witnessed negative growth in total factor productivity. From 2016 to 2020, the EFFCH and SECH of health resource allocation efficiency showed positive growth, with an average annual growth rate of 0.065% and 0.505%, respectively. Technological progress and

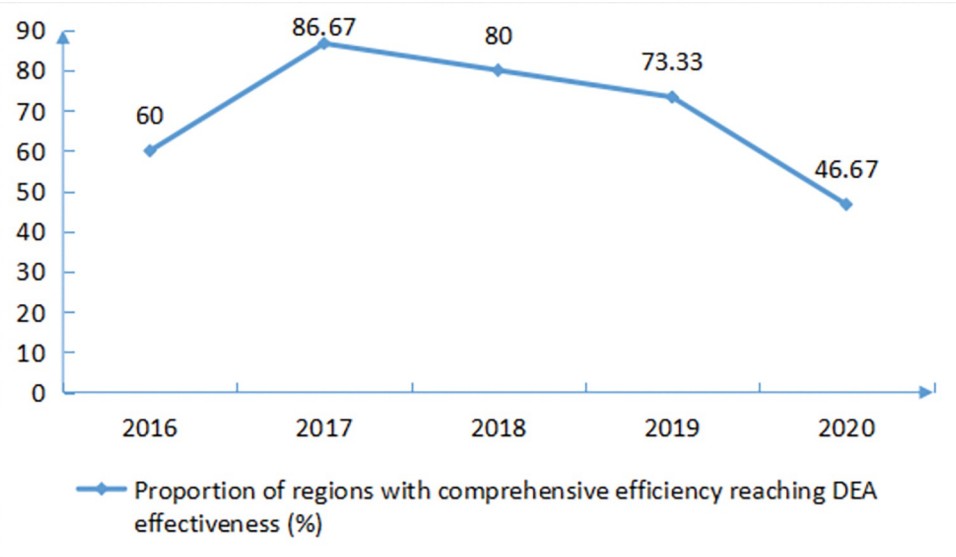

**Fig 1. Proportion of sample areas with DEA effective comprehensive efficiency.**

pure technical efficiency showed negative growth, with an average annual decline of 6.52% and 0.403%, respectively (Table 3).

From 2016 to 2020, only one area in Hainan Province had total factor productivity of medical and health resources greater than 1. Only the technological progress and pure technical efficiency of Haikou City increased synchronously, while the technological progress index of other cities and counties decreased, and the technological progress of Wanning City was the lowest. The technological progress of 7 cities and counties was lower than the average, accounting for 46.67%. The technical efficiency of 8 cities and counties was lower than the average, accounting for 53.33% (Table 4).

## Discussion

With the construction of the Hainan Free Trade Port, the demand for population and health services increases, and the allocation of health resources also increases year by year. However, through the super-efficiency SBM measurement, it is found that the comprehensive efficiency score of the samples is 0.975, which indicates that the efficiency of healthcare resource allocation in Hainan Province is not high and needs to be further improved. 2017 was relatively good, but the comprehensive efficiency from 2019 to 2020 was all less than 1, and there was a downward trend. The main reasons for the decline may be these: First, in 2017, it was emphasized in the Key Tasks of Deepening the Reform of the Medical and Health System in 2017

**Table 3. Malmquist index of medical and health resource allocation in Hainan Province from 2016 to 2020.**

| Time interval | Technical efficiency (EFFCH) | Technical progress (TECH) | Pure technical efficiency (PECH) | Scale efficiency (SECH) | Total factor productivity (TFP) |
|---|---|---|---|---|---|
| **2016–2017** | 1.108 | 0.993 | 1.086 | 1.030 | 1.099 |
| **2017–2018** | 0.988 | 0.950 | 1.002 | 0.987 | 0.936 |
| **2018–2019** | 0.961 | 0.974 | 0.977 | 0.985 | 0.935 |
| **2019–2020** | 0.946 | 0.822 | 0.919 | 1.019 | 0.766 |
| **Mean** | 1.001 | 0.935 | 0.996 | 1.005 | 0.934 |

**Table 4. Malmquist index of medical and health resource allocation of cities and counties in Hainan Province from 2016 to 2020.**

| Region | Technical efficiency (EFFCH) | Technical progress (TECH) | Pure technical efficiency (PECH) | Scale efficiency (SECH) | Total factor productivity (TFP) |
|---|---|---|---|---|---|
| Haikou City | 0.977 | 1.006 | 1.046 | 0.980 | 0.981 |
| Sanya City | 1.066 | 0.907 | 1.028 | 1.021 | 0.962 |
| Wuzhishan City | 1.019 | 0.971 | 1.017 | 1.002 | 0.991 |
| Wenchang City | 0.972 | 0.902 | 0.963 | 1.001 | 0.868 |
| Qionghai City | 0.971 | 0.926 | 0.992 | 0.977 | 0.909 |
| Wanning City | 0.880 | 0.865 | 0.896 | 0.975 | 0.758 |
| Ding 'an County | 0.924 | 0.944 | 0.925 | 0.999 | 0.874 |
| Tunchang County | 0.999 | 0.914 | 1.001 | 0.999 | 0.911 |
| Chengmai County | 1.251 | 0.976 | 1.114 | 1.103 | 1.225 |
| Danzhou City | 1.023 | 0.913 | 1.001 | 1.019 | 0.900 |
| Dongfang City | 0.950 | 0.877 | 0.916 | 1.048 | 0.830 |
| Baoting County | 1.004 | 0.964 | 1.038 | 0.992 | 0.968 |
| Lingshui County | 0.899 | 0.963 | 0.901 | 1.000 | 0.871 |
| Baisha County | 1.021 | 0.952 | 1.045 | 0.966 | 0.971 |
| Changjiang County | 1.054 | 0.942 | 1.059 | 0.995 | 0.993 |
| Mean | 1.001 | 0.935 | 0.996 | 1.005 | 0.934 |

that the year 2017 was a key year for China to form a systematic framework of the basic medical and health system and complete the phased objectives of the medical reform. Therefore, China promulgated a number of medical reform policies, such as the cancellation of the drug price addition policy, the comprehensive launch of the medical consortium pilot work, and the implementation of a hierarchical diagnosis and treatment system. The medical and health service system in our country has broken the inherent mode under the effects of this series of medical reforms. During the initial period of the implementation of the new and old policies, it has exerted influence on the normal operation of health undertakings. Second, it may have something to do with the construction of free trade ports in Hainan Province starting in 2018. Hainan has given priority to institutional integration and innovation in its development. Various preferential policies have been formulated and introduced to attract capital, talent, high-end equipment, and other resources to Hainan, breaking the state of self-sufficiency in the past. With the deepening of the reform of the medical system, medical and health resources are being restructured and adjusted, and there is a phenomenon of underutilization and reasonable operation, resulting in a decline in efficiency instead of improvement. This also reflects from another side that no matter how much health resources are invested, if the resources are not reasonably and fully utilized, the efficiency of resource allocation will decline and the output will be poor. Third, due to the impact of the global COVID-19 epidemic in 2019–2020, health resources have been urgently deployed, including equipment, beds, personnel and other resources, focusing on the allocation of the most affected areas. More beds were transferred to treat COVID-19 patients. Medical resources are diverted to emergency response to the epidemic, thus affecting the original efficiency.

Malmquist index shows that technical efficiency, pure technical efficiency, and scale efficiency have positive growth in 2016–2017, indicating that technical efficiency contributes to the improvement of total factor productivity. The decline of total factor productivity from 2017 to 2020 is influenced by both scale efficiency and technological progress and is greatly influenced by technological regress. This is consistent with the studies of Cai Zihan (2022) [16]

and Tian Haoguo (2022) [17]. The reason for this is that, on the one hand, Hainan Province is unable to improve healthcare output through scientific and technological progress and innovation. On the other hand, the current investment in healthcare by the government of Hainan Province is too focused on hardware elements, and to a certain extent, it has neglected the improvement of healthcare technology, the quality of healthcare services, and the quality of health personnel. Of course, hardware inputs are essential, especially financial inputs. But these human resources inputs in health care are also very important. Medical and healthcare human resources are the core of the development of healthcare, which is a key factor. Human resources for health care have a significant impact on the development of health care in Hainan's free trade port and are important for the provision of high-quality health services, the promotion of medical tourism, and the protection of health security. Adequate human resources for health can provide a wider range and quality of health services and ensure that residents have timely access to medical and healthcare services. The development of a free trade port in Hainan implies the possibility of population mobility and growth, thus requiring sufficient human resources for health to meet the growing demand for health services. As a province with a highly developed tourism industry, Hainan has sufficient human resources for health to provide a high level of medical services and attract patients to the province. At the same time, Hainan's free trade port will require stronger human resources to ensure health safety and epidemic prevention and control.

The obvious differences among regions also reflect the unbalanced efficiency of medical resource allocation. The economically developed and populous region of Hainan Province has relatively high health resource allocation efficiency. Haikou City is the capital city of Hainan Province, and its annual gross domestic product (GDP) and population are the largest. Table 2 from the study also found that Haikou City has the highest overall efficiency. This may be related to the government's deepening reform of the healthcare system, adequate investment, and management. Economically developed regions usually have higher levels of investment in health resources, including medical facilities, equipment, and personnel, and such investment can provide better allocation and utilization of health resources and improve efficiency. They usually have more advanced medical technologies and management experience, which can help optimize health resource allocation and improve efficiency. For example, the application of 5G technology and electronic medical records can facilitate hierarchical treatment and improve the efficiency of healthcare organizations. The research and development of new drugs and improvements in surgical techniques can improve the efficiency of health resource allocation. Developed regions usually have better health policies and can receive greater policy concessions. For example, Haikou has many targeted health system integration and innovation measures. Many pilot measures will be implemented first in Haikou City. Areas with large populations usually have higher demand for health services, which prompts the government and healthcare organizations to increase investment in health. In addition, economically developed and populous areas are likely to be better educated and more aware of health care, which also drives the efficient use of health resources. This is consistent with Hang Ranfeng's study of health resource allocation efficiency in Jiangsu Province (2022) [18] and Huang Lili's study of health resource allocation efficiency in Hunan Province (2022) [19]. However, from the perspective of total factor productivity, Haikou City is restricted by scale efficiency, indicating that the medical scale of Haikou City should be properly expanded. Inefficient cities and counties with large areas and small populations, there are redundant inputs and insufficient output, and it is difficult to make full use of health resources. Therefore, it is necessary for each district and county government to reform the management mode, optimize the medical service and improve the service utilization efficiency, rather than compress the input.

## Conclusions and suggestions

The study found that the allocation efficiency of medical and health resources in Hainan Province was generally poor from 2016 to 2020, showing a downward trend. There are also significant regional differences. It is influenced by the double factors of scale efficiency and technological progress, among which technological progress is the key factor. The economically developed areas with large populations in Hainan Province are relatively efficient.

Improving the efficiency of the allocation of medical and health resources is an important guarantee for promoting the development of the health industry of the Hainan Free Trade Port and an inevitable requirement for the high-quality construction of the Hainan Free Trade Port. According to the conclusions of the paper, in order to improve the efficiency of medical and health resources in Hainan Free Trade Port, Hainan Province should rationalize the allocation of medical hardware and software resources based on demographic, geographic, and demand factors, accelerate the promotion of hierarchical diagnosis and treatment and "Internet+" medical care, and promote the expansion and balanced allocation of high-quality medical resources to improve efficiency. In addition, Hainan Province should accelerate the introduction and cultivation of health professionals, focusing on the improvement of medical and health technologies, the quality of medical and health services, and the quality of medical and health personnel.

Hainan Province is a window for Southeast Asian countries to learn more about China. The construction of free trade ports with Chinese characteristics is not only of great significance to China but also conducive to the Belt and Road Initiative and the development of Southeast Asian countries. Under the policy of free trade port with Chinese characteristics, this study adopts the super-efficiency SBM—Malmquist model to analyze the allocation of health resources, which can make up for the shortcomings of traditional data envelopment analysis. At the same time, to some extent, it complicates the research blank of domestic and foreign scholars on the medical and health construction of the Hainan Free Trade Port. It not only provides certain reference values for the health cause of Hainan Province but also attracts the attention of other places in China and even global health resources. However, this study has some limitations. First, the impact of environmental factors on input-output indicators is not further explored. This may lead to a discrepancy between the estimated results and the actual situation. Factors such as economy, population, policy, health personnel, geographical location, transport and education can have a significant impact on the efficiency of health resource allocation. Regions with relatively good environmental factors will have more policy support, greater demand for health care and more economic and financial input. This results in a relatively high efficiency of healthcare resource allocation. Secondly, there are some missing data in individual regions that cannot be fully covered. This may result in not reflecting the overall situation of health resource allocation efficiency in Hainan Province, making the sample incomplete or biased, which may cause errors in the conclusions and affect the robustness of the results. The focus of the later project is to study the influencing factors and the degree of influence on the allocation efficiency of health resources in Hainan Province, while the missing data will be made up by in-depth research or sensitivity analysis, etc. to ensure that the data are more robust.

## Acknowledgments

We gratefully acknowledge the anonymous reviewers for their valuable comments and suggestions. We are grateful to all who supported our research and contributed to this study.

## Author Contributions

**Conceptualization:** Yanhua Gong, Dong Ma.

**Data curation:** Yanhua Gong, Dong Ma, Wen Feng.

**Formal analysis:** Yanhua Gong, Dong Ma, Wen Feng.

**Investigation:** Yanhua Gong, Dong Ma.

**Methodology:** Yanhua Gong, Dong Ma.

**Project administration:** Yanhua Gong, Dong Ma.

**Writing – original draft:** Yanhua Gong, Dong Ma.

**Writing – review & editing:** Yanhua Gong, Dong Ma.

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
