## [Decision Letter · Decision Letter 0]

5 Jul 2023

PONE-D-23-10563Study on the allocation efficiency of medical and health resources in Hainan Province under the construction of Free Trade Port: Based on the super-efficiency SBM — Malmquist modelPLOS ONE

Dear Dr. Gong,

Thank you for submitting your manuscript to PLOS ONE. After careful consideration, we feel that it has merit but does not fully meet PLOS ONE’s publication criteria as it currently stands. Therefore, we invite you to submit a revised version of the manuscript that addresses the points raised during the review process.

 You may find the detailed comments from Reviewers below, however I would like to draw your attention to few important issues, addressing which will improve the quality of your manuscript. Firstly, it will be beneficial to address the context of the problem in more detail. One of the Reviewers pointed out the environmental limitations in resource allocation this field, the other one human resources and the impact of Hainan being a free port on healthcare. These are only examples - addressing similar issues will definitly make your study more understandable for international public. Secondly, the case of missing data is also of some concern and in my opinion you need to discuss how it influences the study robustness. Both Reviewers also indicate other issues that need clarification within the manuscript - as mentioned, you may find the whole list below. Please submit your revised manuscript by Aug 14 2023 11:59PM. If you will need more time than this to complete your revisions, please reply to this message or contact the journal office at plosone@plos.org. Please include the following items when submitting your revised manuscript:A rebuttal letter that responds to each point raised by the academic editor and reviewer(s). You should upload this letter as a separate file labeled 'Response to Reviewers'.A marked-up copy of your manuscript that highlights changes made to the original version. You should upload this as a separate file labeled 'Revised Manuscript with Track Changes'.An unmarked version of your revised paper without tracked changes. You should upload this as a separate file labeled 'Manuscript'.If applicable, we recommend that you deposit your laboratory protocols in protocols.io to enhance the reproducibility of your results. Protocols.io assigns your protocol its own identifier (DOI) so that it can be cited independently in the future. For instructions see: https://journals.plos.org/plosone/s/submission-guidelines#loc-laboratory-protocols. Additionally, PLOS ONE offers an option for publishing peer-reviewed Lab Protocol articles, which describe protocols hosted on protocols.io. Read more information on sharing protocols at https://plos.org/protocols?utm_medium=editorial-email&utm_source=authorletters&utm_campaign=protocols.

We look forward to receiving your revised manuscript.

Kind regards,

Agata Sielska, Ph.D.

Academic Editor

PLOS ONE

Journal Requirements:

“The author(s) received no specific funding for this work.

Reviewers' comments:

Reviewer's Responses to Questions

**Comments to the Author**

1. Is the manuscript technically sound, and do the data support the conclusions?

Reviewer #1: Yes

Reviewer #2: Yes

2. Has the statistical analysis been performed appropriately and rigorously? 

Reviewer #1: No

Reviewer #2: Yes

3. Have the authors made all data underlying the findings in their manuscript fully available?

Reviewer #1: Yes

Reviewer #2: Yes

4. Is the manuscript presented in an intelligible fashion and written in standard English?

Reviewer #1: Yes

Reviewer #2: Yes

5. Review Comments to the Author

Reviewer #1: The manuscript titled "Equity and Efficiency of Medical and Health Resource Allocation: A Case Study of Hainan Province" provides valuable insights into the allocation of medical and health resources in Hainan Province, China. The authors effectively address the significance of equity and efficiency in resource allocation, highlighting the impact on sustainable and high-quality development of health services. The study aims to improve resource allocation efficiency and provide references for other regions.

This manuscript contributes to the literature on health resource allocation and offers meaningful recommendations for optimizing resource utilization in Hainan Province. Overall, the manuscript is well-written and presents a systematic analysis of medical and health resource allocation in Hainan Province. However, it is important to note that the manuscript could benefit from further discussion on the limitations of the study, particularly regarding the influence of environmental factors on resource allocation and the missing data in some regions. Addressing these limitations would enhance the robustness of the research.

In addition, the authors should clarify the following questions.

How did the study measure the comprehensive efficiency of medical and health resource allocation in Hainan Province?

What specific factors contribute to the decline in the allocation efficiency of medical and health resources in Hainan Province, particularly during the years 2019 and 2020?

Are there any specific policies or strategies mentioned in the manuscript that address the issues identified in the allocation efficiency of medical and health resources in Hainan Province?

What evidence or examples are provided to support the claim that economically developed areas with large populations have relatively higher allocation efficiency of health resources in Hainan Province?

Reviewer #2: The article discusses the most important issues for the functioning of the health care system, i.e. the efficiency of employment of production factors. Comments on the content of the article are presented in the following points in accordance with the order of issues in the article:

-/ in the description of the province, the number of population should be given,

1/ the authors focus on human resources, and is it not worth considering other resources, there is also a budget constraint that affects the efficiency of management, and thus also clinical effectiveness,

2/ it is incomprehensible to refer to a free port, what impact does it have on the functioning of health care? Are there any dependencies?

3/ introducing solutions that work well in health care in one country does not lead to success in another country. Therefore, it is incomprehensible to refer to Singapore. The differences between the Chinese province and Singapore are so large that the solutions from Singapore cannot be expected to have positive effects in China. Thorough research is definitely needed

4/ the introduction should include a more detailed description of the study itself,

5/ institutions appear as input indicators. What institutions are we talking about? Are they supposed to be service providers?

6/ policy conclusions are vaguely worded.

6. PLOS authors have the option to publish the peer review history of their article (what does this mean?). If published, this will include your full peer review and any attached files.

Reviewer #1: No

Reviewer #2: No

---

## [Author Response · Author response to Decision Letter 0]

17 Jul 2023

Dear Reviewers:

Thank you very much for your comments and professional advice on the manuscript (Title: Study on the allocation efficiency of medical and health resources in Hainan Province under the construction of Free Trade Port: Based on the super-efficiency SBM — Malmquist model. Number: PONE-D-23-10563). These opinions are helpful to improve the academic and rigorous nature of the manuscript. In this regard, we hold a serious and steadfast learning attitude. According to your suggestions, we have carefully revised the manuscript and adjusted and polished some of the language. We hope the manuscript can be better. We have not changed much in the overall framework, but we have changed the "Conclusion and policy implications" section to "Conclusions and suggestions" because the term is more consistent with our manuscript. In addition, the "Introduction" section, the "Discussion" section, the "Conclusions and suggestions" section and the "limitations" section have also been greatly revised, which are the focus of attention of experts. In a marked revision manuscript, we have highlighted your proposed revision in yellow and the problems we found and the appropriate changes to the original text in red. Also, in this reply, we have highlighted some of the content in gray so that you can look more clear. Specific modification details are as follows:

Reviewer #1:

Thank you very much for your recognition of the manuscript and your professional suggestions. The following are our amendments and replies.

1.The manuscript titled "Equity and Efficiency of Medical and Health Resource Allocation: A Case Study of Hainan Province" provides valuable insights into the allocation of medical and health resources in Hainan Province, China. The authors effectively address the significance of equity and efficiency in resource allocation, highlighting the impact on sustainable and high-quality development of health services. The study aims to improve resource allocation efficiency and provide references for other regions. This manuscript contributes to the literature on health resource allocation and offers meaningful recommendations for optimizing resource utilization in Hainan Province. Overall, the manuscript is well-written and presents a systematic analysis of medical and health resource allocation in Hainan Province. However, it is important to note that the manuscript could benefit from further discussion on the limitations of the study, particularly regarding the influence of environmental factors on resource allocation and the missing data in some regions. Addressing these limitations would enhance the robustness of the research.

The author's answer：Thank you very much for your recognition of the manuscript and your professional suggestions. Based on your suggestions, we have had a more specific discussion about the limitations. (Lines 371-384) The specific changes are as follows:

Original text：However, there are some limitations in this study, which do not further investigate the influence of environmental factors on the input and output index. There are some missing data in individual regions, which cannot be fully covered. It is hoped to improve it in the later period.

The modified：However, this study has some limitations. First, the impact of environmental factors on input-output indicators is not further explored. This may lead to a discrepancy between the estimated results and the actual situation. Factors such as economy, population, policy, health personnel, geographical location, transport and education can have a significant impact on the efficiency of health resource allocation. Regions with relatively good environmental factors will have more policy support, greater demand for health care and more economic and financial input. This results in a relatively high efficiency of healthcare resource allocation. Secondly, there are some missing data in individual regions that cannot be fully covered. This may result in not reflecting the overall situation of health resource allocation efficiency in Hainan Province, making the sample incomplete or biased, which may cause errors in the conclusions and affect the robustness of the results. The focus of the later project is to study the influencing factors and the degree of influence on the allocation efficiency of health resources in Hainan Province, while the missing data will be made up by in-depth research or sensitivity analysis, etc. to ensure that the data are more robust.

2.How did the study measure the comprehensive efficiency of medical and health resource allocation in Hainan Province?

The author's answer：We are sorry that we only wrote about the measure of dynamic efficiency and total factor productivity in "Data and methods" in the paper, but did not explicitly write how to measure comprehensive efficiency. Thank you for spotting this. We have added this part to the "Super-efficiency SBM" section of the text. (Lines 168-172) Specifically add as follows:

Added content: According to the super-efficiency SBM model, ρ < 1 indicates that the comprehensive efficiency of resource allocation is in a relatively invalid state and the efficiency is low. ρ ≥ 1 indicates that the comprehensive efficiency of resource allocation is in a relatively effective state and the efficiency is high. It is generally believed that the larger the comprehensive efficiency value, the higher the efficiency of resource allocation, and the better the level.

3.What specific factors contribute to the decline in the allocation efficiency of medical and health resources in Hainan Province, particularly during the years 2019 and 2020?

The author's answer：The factors leading to the decline in the efficiency of medical and health resource allocation in Hainan Province include policy, economic development, population, geography and transportation. The reasons are also mentioned in the "Discussion" section of this manuscript (lines 255-278), but may need to be more specific. Therefore, we have made a more detailed supplement to the first paragraph of "Discussion" for readers to understand. Mainly, the second and third reasons are supplemented in detail. Specific modifications are as follows:

Original text：First, in 2017, it was emphasized in the Key Tasks of Deepening the Reform of the Medical and Health System in 2017 that the year 2017 was a key year for China to form a systematic framework of the basic medical and health system and complete the phased objectives of the medical reform. Therefore, China promulgated a number of medical reform policies, such as the cancellation of the drug price addition policy, the comprehensive launch of the medical consortium pilot work, and the implementation of a hierarchical diagnosis and treatment system. The medical and health service system in our country has broken the inherent mode under the effects of this series of medical reforms. During the initial period of the implementation of the new and old policies, it has exerted influence on the normal operation of health undertakings. Second, it may have something to do with the construction of free trade ports in Hainan Province starting in 2018. Hainan has given priority to institutional integration and innovation in its development. Various preferential policies have been formulated and introduced to attract capital, talent, high-end equipment, and other resources to Hainan, breaking the state of self-sufficiency in the past. The reform of the medical system has been deepened, and the medical and health resources are being restructured. Third, the impact of the global COVID-19 pandemic has led to low efficiency of medical resources.

The modified：First, in 2017, it was emphasized in the Key Tasks of Deepening the Reform of the Medical and Health System in 2017 that the year 2017 was a key year for China to form a systematic framework of the basic medical and health system and complete the phased objectives of the medical reform. Therefore, China promulgated a number of medical reform policies, such as the cancellation of the drug price addition policy, the comprehensive launch of the medical consortium pilot work, and the implementation of a hierarchical diagnosis and treatment system. The medical and health service system in our country has broken the inherent mode under the effects of this series of medical reforms. During the initial period of the implementation of the new and old policies, it has exerted influence on the normal operation of health undertakings. Second, it may have something to do with the construction of free trade ports in Hainan Province starting in 2018. Hainan has given priority to institutional integration and innovation in its development. Various preferential policies have been formulated and introduced to attract capital, talent, high-end equipment, and other resources to Hainan, breaking the state of self-sufficiency in the past. With the deepening of the reform of the medical system, medical and health resources are being restructured and adjusted, and there is a phenomenon of underutilization and reasonable operation, resulting in a decline in efficiency instead of improvement. This also reflects from another side that no matter how much health resources are invested, if the resources are not reasonably and fully utilized, the efficiency of resource allocation will decline and the output will be poor. Third, due to the impact of the global COVID-19 epidemic in 2019–2020, health resources have been urgently deployed, including equipment, beds, personnel and other resources, focusing on the allocation of the most affected areas. More beds were transferred to treat COVID-19 patients. Medical resources are diverted to emergency response to the epidemic, thus affecting the original efficiency.

4.Are there any specific policies or strategies mentioned in the manuscript that address the issues identified in the allocation efficiency of medical and health resources in Hainan Province?

The author's answer：Yes, in view of the problems existing in the efficiency of medical and health resource allocation in Hainan Province, it is important to reflect in the "14th Five-Year Plan for Hygiene and Health of Hainan Province", "14th Five-Year Plan for Grassroots Hygiene and Health of Hainan Province", "14th Five-Year Plan for Digital Health Development of Hainan Province" and other policies. The article lists specific strategies to improve efficiency in the "Conclusions and Suggestions" part. But that may not be specific enough. Based on the actual situation in Hainan Province, we modified the strategy in more detail to make it more operable (lines 313-359). The basic framework and keywords have not changed more, but the relevant content has been expanded to be more specific. It mainly includes the system innovation, the balance of medical and health resources, the cultivation and introduction of health talents, and the exchange and cooperation at home and abroad. The details are as follows:

Original text：First, Hainan should continue to take institutional integration and innovation as the "starting point" to improve the ability of modern governance of health services. Hainan Province should accelerate the implementation and optimization of policies, accelerate the progress of medical reform, the development of Boao International Medical Tourism Pilot Zone and the development of the healthcare industry, attach importance to medical innovation and achievement transformation, and promote the health industry to enter the fast track of high-quality development steadily. Second, Hainan should balance the distribution and optimization of health resources, and speed up the process of promoting graded diagnosis and treatment and "Internet Plus" medical treatment. Strengthen supervision and management, appropriately expand the scale of medical services in big cities, and for inefficient cities and counties, and improve the efficiency of service utilization by means of technology introduction, graded diagnosis and treatment, and "Internet +" medical treatment while ensuring adequate input. To speed up the development of Haikou, Sanya, and Danyang economic circles, to form a new pattern of development, and to improve medical consortium, hospital groups, and other models of division of labor and collaboration. Third, strengthen cooperation and exchanges and continue to upgrade hardware elements. To maintain close contact with Singapore, Shanghai, Beijing, Guangdong Province, and other places and strengthen technical exchanges, and improve the level of medical technology and service efficiency. First-class medical institutions and research centers at home and abroad are introduced to Hainan to achieve joint construction, governance, and sharing, and services for the construction of the Hainan Free trade Port. Fourth, accelerate the introduction and training of health professionals to enhance the soft power of health. On the one hand, in order to adapt to the development of the Hainan Free Trade Port, it needs to increase the total number of health professionals. On the other hand, relying on favorable policies, Hainan will focus on introducing and training high-level health professionals, mastering the core medical technology, and creating high-end medical services in the world. In addition, the policy should be appropriately tilted to the community level to speed up the construction of health professionals at the community level and guarantee their treatment. Improving the allocation efficiency of medical and health resources is an important guarantee for promoting the development of the health industry of the Hainan Free Trade Port, and an inevitable requirement for the high-quality construction of the Hainan Free Trade Port.

The modified：First, Hainan should continue to take institutional integration and innovation as the "starting point" to improve the ability of modern governance of health services. The "14th Five-Year Plan for Hygiene and Health of Hainan Province" issued by Hainan Province in 2021 emphasizes that Hainan must accelerate the expansion and balanced distribution of high-quality medical and health resources and improve the allocation efficiency of medical and health resources. The Hainan provincial government should further implement and optimize policies, accelerate the process of medical reform, improve detailed medical and health resource planning, and rationally allocate medical and health hardware and software resources according to population, geography, demand, and other factors in order to improve efficiency. Accelerate the development of key health projects in Hainan Province. For example, promote the implementation of the global "One Health" demonstration project in Hainan Province through publicity and improvement of the medical and prevention integration governance model; Through the high-level equipment of hardware and software facilities, increase the degree of policy openness, pay attention to medical innovation and achievement transformation, and promote the process of the Boao Lecheng International Medical Tourism Pilot Zone; Based on the unique natural environment of Hainan Province, it is necessary to promote the rational development of the healthcare industry. 

Second, Hainan should balance the layout and optimize the allocation of health resources, and speed up the process of promoting graded diagnosis and treatment and "Internet Plus" medical treatment. Strengthen supervision and management, appropriately expand the scale of medical services in big cities, and for inefficient cities and counties, and improve the efficiency of service utilization by means of technology introduction, graded diagnosis and treatment, and "Internet +" medical treatment while ensuring adequate input. Speed up the building of economic circles in Haikou, Sanya, and Dan Ocean, form a point with a circle and a circle with an area, promote the formation of a new development pattern, accelerate the implementation of the relevant implementation opinions of Hainan Province's grid-tight medical and health service system, improve the medical consortium, hospital group, and other division of labor and cooperation models, and comprehensively improve diagnosis and treatment efficiency and resource utilization efficiency. 

Third, accelerate the introduction and training of health professionals, expand the total number, and improve the quality. On the one hand, the establishment of first-class medical colleges, especially nursing colleges. Independently cultivate and stabilize health professionals. On the other hand, Hainan Province can rely on favorable policies to improve the salary of health professionals, innovate the post and staffing system, focus on introducing and training high-level health professionals, master the core medical technology, improve the level of medical technology, and create high-end medical services in the world. In addition, the policy should be appropriately tilted to the grass-roots level, accelerate the construction of the grass-roots health personnel team, innovate the assessment system of grass-roots health personnel, pay more attention to the improvement of their post-competence, service attitude, diagnosis and treatment level, and ensure the quality and quantity of grass-roots health personnel. 

Fourth, strengthen international cooperation and exchanges to promote the sharing of medical and health resources. To maintain close contact with Beijing, Shanghai, Guangdong Province, and other places to strengthen technical exchanges and improve the level of medical technology and service efficiency. The introduction of domestic and foreign first-class medical institutions and research centers in Hainan to achieve co-construction, co-governance, and sharing to serve the construction of the Hainan Free Trade Port.

5.What evidence or examples are provided to support the claim that economically developed areas with large populations have relatively higher allocation efficiency of health resources in Hainan Province?

The author's answer：Haikou City is a relatively developed area in Hainan Province. Haikou City is the capital city of Hainan Province, and its annual GDP and population are the largest. Table 2 also shows that the comprehensive efficiency of Haikou City is the highest. To solve this problem, we can find some reasons in the following:

(1)High investment. Economically developed areas usually have a higher level of investment in health resources, including investment in medical facilities, equipment and personnel, which can provide better allocation and utilization of health resources and improve efficiency.

(2)Advanced technology and management experience. Economically developed regions usually have more advanced medical technology and management experience, which helps optimize the allocation of medical resources and improve efficiency. For example, the application of 5G technology and electronic medical records can promote hierarchical diagnosis and treatment and improve the efficiency of healthcare institutions. The research and development of new drugs and the improvement of surgical techniques can improve the efficiency of health resource allocation.

(3) Policy support. Developed regions usually have better health policies and can get greater policy preferences. For example, Haikou City has many targeted innovative measures of health system integration. Many of the pilot measures will be implemented first in Haikou.

(4)Patient needs. Areas with large populations usually have a higher demand for health services, which encourages governments and medical institutions to increase health investment. In addition, economically developed areas with large populations are likely to be better educated and more health-conscious, which will also promote efficient use of health resources.

Reviewer #2:

Thank you very much for your recognition of the manuscript and your professional suggestions. The following are our amendments and replies.

1.In the description of the province, the number of population should be given.

The author's answer：Thank you very much for your reminding. Yes, we quite agree with your suggestion. And we have to make changes. After consideration, we added the population of Hainan in 2020 to the description of the province in the "Introduction" section. (Line 104)

Add content:and the number of permanent residents in 2020 is 10,123,400 people.

2.The authors focus on human resources, and is it not worth considering other resources, there is also a budget constraint that affects the efficiency of management, and thus also clinical effectiveness.

The author's answer：Medical and health human resources are very important, they are the core and the key factor of health development. Medical and health human resources have an important impact on the development of health undertakings in Hainan Free Trade Port. Specific aspects may be as follows:

(1)Access to health services. Adequate human resources for health can provide a wider range of high-quality health services and ensure that the population has timely access to medical and health services. The development of Hainan Free Trade Port means the possibility of population flow and growth, so sufficient health human resources are needed to meet the growing demand for health services.

(2)Health status of the population. Whether the health human resources are sufficient or not will directly affect the health status of residents in Hainan Free Trade Port. Adequate doctors, nurses and other medical professionals can provide comprehensive medical services, prevent and control diseases, and improve the health of residents.

(3)Development of medical tourism. The Hainan Free Trade Port has developed medical tourism to attract patients from home and abroad to seek medical services in Hainan. This requires Hainan to have sufficient health human resources to provide high-level medical services and attract patients to seek medical treatment.

(4)Health security and epidemic prevention and control: Hainan Free Trade Port needs to have stronger health human resources to ensure health security and epidemic prevention and control. To ensure the health and safety of residents and visitors.

In short, health human resources have an important impact on the development of Hainan Free Trade Port, and are of great significance for providing high-quality health services, promoting the development of medical tourism, and ensuring health safety. Therefore, Hainan needs to pay attention to the training and allocation of health human resources to support the rapid development of free trade port.

But this does not mean focusing only on the health workforce, other resources are also very important. When selecting indicators in this study, there are also indicators related to equipment and facilities, (Lines 134-135) and the influence of personnel, capital and material will also be mentioned in the "Discussion" sections.(Lines 266-268) However, it is undeniable that this study does involve less capital (budget) input. In the index selection, capital input was not selected as one of the input indicators. This is because, considering that financial resources are almost transformed into human and material resources, if financial resources are added as input indicators, duplication may occur. And data on capital are harder to come by. There is no denying that capital is indeed a very important influencing factor, and we will focus on this factor in the later study of influencing factors.

3. It is incomprehensible to refer to a free port, what impact does it have on the functioning of health care? Are there any dependencies?

The author's answer：Hainan Province is currently the only free trade port with Chinese characteristics in mainland China. Under the background of free trade port, Hainan Province will have more development opportunities. The Hainan Free Trade Port may have the following impacts on the operation of health care:

(1)Provide more medical resources. The free trade port policy is expected to attract more domestic and foreign medical institutions and professionals to enter Hainan, increase the supply of medical resources, and improve the level and quality of medical services.

(2) Promote medical technology innovation. The free trade port policy encourages scientific and technological innovation and intellectual property protection, which helps attract medical technology and pharmaceutical innovation enterprises to enter Hainan and promote the development and application of medical technology.

(3) Improve the quality of medical services. The construction of the free trade port will promote the competition between medical institutions and medical personnel, forcing them to improve the quality of medical services and provide better medical experience to patients.

(4) Promote the development of medical tourism. The free trade port policy is expected to attract more international patients to Hainan for medical treatment, promote the development of medical tourism, and increase the demand for medical services and economic income in Hainan.

(5) Optimize the medical management system: The construction of the free trade port may bring about the reform and innovation of the medical management system, promote the standardization and modernization of medical institutions, and improve the operation efficiency and management level.

4.Introducing solutions that work well in health care in one country does not lead to success in another country. Therefore, it is incomprehensible to refer to Singapore. The differences between the Chinese province and Singapore are so large that the solutions from Singapore cannot be expected to have positive effects in China. Thorough research is definitely needed.

The author's answer：Thank you very much for your advice. We agree with your statement. Every region is different. This article mentions Singapore in both the "Introduction" and "Conclusions and suggestions" sections. After careful review and consideration, we have revised the article. In the "Introduction", Singapore's experience is mainly derived from a piece of literature, and it may be more suitable for this paper to put this literature into domestic and foreign research, rather than taking Singapore out alone. (Lines 79-82) In the section of "Conclusions and suggestions", the reference to Singapore's experience is removed. (Line 355-356) It is not because Singapore has no experience to learn from, but we agree with your opinion. At the same time, we consider that most of Singapore's medical institutions are private, and there is a big difference between Singapore and China. Therefore, the experience of other excellent regions in China is more in line with the development trend of Hainan Province.

The world is open and inclusive in many ways. Singapore has been relatively successful in the process of building a free trade port. Singapore has done better in primary health care and community services. If we have the opportunity, we will thoroughly study its specific practices and see if it can bring some inspiration to the construction of Hainan Free Trade Port. We really appreciate your advice.

Original text：In Sayem Ahmed's study on the efficiency of health systems in Asia, Singapore's efficiency is effective, indicating that it makes full use of health system resources.[4] Singapore is also a free trade port, which is worth learning from Hainan Free Trade Port.

The modified：Sayem Ahmed ( 2019 ) [4] used the DEA, Tobit model, and smooth bootstrap model to evaluate the technical efficiency of health systems in Asian countries and found that up to 91.3% of countries had inefficient utilization of health resources. (Lines 79-82)

Original text：strengthen cooperation and exchanges and continue to upgrade hardware elements. To maintain close contact with Singapore, Shanghai, Beijing, Guangdong Province, and other places and strengthen technical exchanges,

The modified：To maintain close contact with Beijing, Shanghai, Guangdong Province, and other places to strengthen technical exchanges and improve the level of medical technology and service efficiency. (Line 355-356)

5.The introduction should include a more detailed description of the study itself.

The author's answer：Thank you very much for your advice. It is true that the "Introduction" part of this study is not good enough. We revised it in detail to make it more specific and clear. In the "Introduction" part, we focus on the research theme, writing the research background, significance, domestic and foreign research, methodological research and research purpose. (Lines 46-123) The specific changes are as follows:

Original text：The contradiction between the finiteness of health resources and the infinite demand of people is prominent.[1] How to make more effective use of the limited health resources, meet the needs of people at different levels of health services, optimize the distribution of health resources and boost the efficiency of use is a common problem faced by the global medical and health reform.[1, 2] According to the "Healthy China 2030" Plan's Outline in 2016, by 2030, China's health institutions and systems will be improved, the development of the health sector will be more coordinated, healthy lifestyles will be popularized, and the quality and guarantee of health services will continue to improve. The important premise of improving health levels and promoting health equity is the rational allocation and efficient utilization of medical and health resources. Hainan Province is the only free trade port in China and is also known as a "test bed for national reform and opening up ", which means that the successful experience of Hainan's reforms may be extended to other parts of China, and its construction is of great significance to China.[3] In Sayem Ahmed's study on the efficiency of health systems in Asia, Singapore's efficiency is effective, indicating that it makes full use of health system resources.[4] Singapore is also a free trade port, which is worth learning from Hainan Free Trade Port. 

Hainan Province (108°37 '-- 111°03' east longitude, 18°10 '-- 20°10' north latitude) is located in the southernmost part of China, close to Southeast Asia, with a total land area of 35,400 square kilometers. In 2018, the Party Central Committee decided to support Hainan in gradually exploring and steadily deepening the construction of a free trade port with Chinese characteristics, which has helped the province comprehensively deepen reform. Hainan is committed to building Health Island and Longevity Island. Under the free trade port policy, Hainan has focused on institutional integration innovation and made major breakthroughs in health care. However, with the increase in population, the intensification of aging, and the change of disease spectrum, the current medical and health resources in Hainan Province are obviously in shortage, and there are outstanding problems such as imbalance, inequity, low efficiency, and lack of human resources. In order to effectively cope with future changes, Hainan must speed up the adjustment and optimization of medical and health resources and improve the efficiency of resource utilization, so as to provide an important guarantee for the high-quality development of health services in Hainan and escort the construction of free trade ports. At present, more attention is paid to the ecological environment and trade of Hainan Province at home and abroad, while there are few related types of research on health status, especially those on Hainan Province under the policy of free trade port with Chinese characteristics. In this study, the super-efficiency SBM and Malmquist index model were used to study the allocation efficiency of medical and health resources in Hainan Province during 2016-2020, analyze the current situation, problems, and causes, and put forward reasonable suggestions to provide a reference for scientific health development planning.

The modified：The contradiction between the finiteness of health resources and the infinite demand of people is prominent.[1] How to make more effective use of the limited health resources, meet the needs of people at different levels of health services, optimize the distribution of health resources and boost the efficiency of use is a common problem faced by the global medical and health reform.[1, 2] According to the "Healthy China 2030" Plan's Outline in 2016, by 2030, China's health institutions and systems will be improved, the development of the health sector will be more coordinated, healthy lifestyles will be popularized, and the quality and guarantee of health services will continue to improve. The important premise of improving health levels and promoting health equity is the rational allocation and efficient utilization of medical and health resources. The improvement of the efficiency of the allocation of medical and health resources can not only promote the accessibility and quality of medical services but also play an important role in improving people's living standards and promoting economic growth. Since the implementation of China's new medical reform in 2009, the amount of health resource investment has generally increased, and great results have been achieved.

Foreign studies on the allocation of health resources started earlier, and the UK was the first country to conduct research on the allocation of health resources [3]. In 1976, the Resource Allocation Working Party (RAWP) was established in the United Kingdom to study the allocation of health resources. Research on the allocation of health resources in China began in the 1980s. With the development of the economy, the rapid increase in population, and the intensification of aging, people have put forward higher requirements for medical and health resources and services, and the government has continuously increased its investment in health resources. The number of health institutions, beds, and health technicians is increasing, and the service capacity of medical and health institutions is greatly improved, but the efficiency of medical and health resource allocation is still low, which may lead to increased investment in subsequent health resources but poor output. This not only fails to meet people's growing medical and health needs but also causes serious waste of medical and health resources, resulting in the rapid growth of medical costs, increasing people's medical burden, and restricting the sustainable and high-quality development of health and health undertakings, which is contrary to the ultimate goal of national health and health development.

How to make the best use of health resources is a hot topic for scholars at home and abroad. Data envelopment analysis ( DEA ) is the most mature and extensive method to study the efficiency of the allocation of medical and health resources. For example, Sayem Ahmed (2019) [4] used the DEA, Tobit model, and smooth bootstrap model to evaluate the technical efficiency of health systems in Asian countries and found that up to 91.3% of countries had inefficient utilization of health resources. Lacko R. (2022) [5] uses the DEA model and Malmquist index to compare the efficiency and productivity of EU countries. It is found that although countries joining the EU get a high technical efficiency value, the overall technical efficiency shows a downward trend, and the productivity of Western countries shows stagnation. Xiang Yingru (2020) [6] analyzed the efficiency of provincial health resources in China by using the BCC model of DEA and the Malmquist index. The study pointed out that after the implementation of medical reform in China, resources are still wasted, output is insufficient in some areas, and efficiency is low due to the excessive number of public hospitals and the serious aging of the population. Technological progress is a key factor affecting efficiency. Luo Liangqing (2008) [7] used the DEA model to study the production efficiency of medical and health services in the eastern, central, and western regions of China and found that the production efficiency of medical and health services in China was generally low. It can be seen from many related studies that the traditional DEA model has some limitations when studying the efficiency of the allocation of medical and health resources. Therefore, the method is improved in this article.

The problem of uneven development among different regions in China has always existed. Hainan Province is currently the only free trade port with Chinese characteristics on the Chinese mainland, and is also known as a "test bed for national reform and opening up ", which means that the successful experience of Hainan's reforms may be extended to other parts of China, and its construction is of great significance to China.[8] Hainan Province (108°37 '-- 111°03' east longitude, 18°10 '-- 20°10' north latitude) is located in the southernmost part of China, close to Southeast Asia, with a total land area of 35,400 square kilometers, and the number of permanent residents in 2020 is 10,123,400 people. In 2018, the Party Central Committee decided to support Hainan in gradually exploring and steadily deepening the construction of a free trade port with Chinese characteristics, which has helped the province comprehensively deepen reform. Hainan is committed to building Health Island and Longevity Island. Under the free trade port policy, Hainan has focused on institutional integration innovation and made major breakthroughs in health care. However, with the increase in population, the intensification of aging, and the change of disease spectrum, the current medical and health resources in Hainan Province are obviously in shortage, and there are outstanding problems such as imbalance, inequity, low efficiency, and lack of human resources. In order to effectively cope with future changes, Hainan must speed up the adjustment and optimization of medical and health resources and improve the efficiency of resource utilization, so as to provide an important guarantee for the high-quality development of health services in Hainan and escort the construction of free trade ports. At present, more attention is paid to the ecological environment and trade of Hainan Province at home and abroad, while there are few related types of research on health status, especially those on Hainan Province under the policy of free trade port with Chinese characteristics. In this study, the super-efficiency SBM model and Malmquist index were used to study the allocation efficiency of medical and health resources in Hainan Province during 2016-2020, analyze the current situation, problems, and causes, and put forward reasonable suggestions to provide a reference for scientific health development planning.

6.Institutions appear as input indicators. What institutions are we talking about? Are they supposed to be service providers?

The author's answer：The indexes selected by data envelopment analysis should be representative, deterministic, independent and stable. This study selects indicators on the basis of literature review. As an input indicator, medical and health institutions mainly include hospitals, primary medical and health institutions, professional public health institutions, internet hospitals, and other medical and health institutions in the Hainan Provincial Health Statistics Yearbook. It reflects the overall size of the region's health systems. It is a public place to provide medical and health services and is one of the carriers.

7.Policy conclusions are vaguely worded.

The author's answer：Thank you very much for your advice. We have reorganized the "Conclusions and suggestions" section with a view to making the recommendations more specific and actionable. (Lines 313–359) The basic framework and keywords have not changed much, but the relevant content has been expanded to be more specific. It mainly includes the system innovation, the balance of medical and health resources, the cultivation and introduction of health talents, and the exchange and cooperation at home and abroad. The specific changes are as follows:

Original text：First, Hainan should continue to take institutional integration and innovation as the "starting point" to improve the ability of modern governance of health services. Hainan Province should accelerate the implementation and optimization of policies, accelerate the progress of medical reform, the development of Boao International Medical Tourism Pilot Zone and the development of the healthcare industry, attach importance to medical innovation and achievement transformation, and promote the health industry to enter the fast track of high-quality development steadily. Second, Hainan should balance the distribution and optimization of health resources, and speed up the process of promoting graded diagnosis and treatment and "Internet Plus" medical treatment. Strengthen supervision and management, appropriately expand the scale of medical services in big cities, and for inefficient cities and counties, and improve the efficiency of service utilization by means of technology introduction, graded diagnosis and treatment, and "Internet +" medical treatment while ensuring adequate input. To speed up the development of Haikou, Sanya, and Danyang economic circles, to form a new pattern of development, and to improve medical consortium, hospital groups, and other models of division of labor and collaboration. Third, strengthen cooperation and exchanges and continue to upgrade hardware elements. To maintain close contact with Singapore, Shanghai, Beijing, Guangdong Province, and other places and strengthen technical exchanges, and improve the level of medical technology and service efficiency. First-class medical institutions and research centers at home and abroad are introduced to Hainan to achieve joint construction, governance, and sharing, and services for the construction of the Hainan Free trade Port. Fourth, accelerate the introduction and training of health professionals to enhance the soft power of health. On the one hand, in order to adapt to the development of the Hainan Free Trade Port, it needs to increase the total number of health professionals. On the other hand, relying on favorable policies, Hainan will focus on introducing and training high-level health professionals, mastering the core medical technology, and creating high-end medical services in the world. In addition, the policy should be appropriately tilted to the community level to speed up the construction of health professionals at the community level and guarantee their treatment. Improving the allocation efficiency of medical and health resources is an important guarantee for promoting the development of the health industry of the Hainan Free Trade Port, and an inevitable requirement for the high-quality construction of the Hainan Free Trade Port.

The modified：First, Hainan should continue to take institutional integration and innovation as the "starting point" to improve the ability of modern governance of health services. The "14th Five-Year Plan for Hygiene and Health of Hainan Province" issued by Hainan Province in 2021 emphasizes that Hainan must accelerate the expansion and balanced distribution of high-quality medical and health resources and improve the allocation efficiency of medical and health resources. The Hainan provincial government should further implement and optimize policies, accelerate the process of medical reform, improve detailed medical and health resource planning, and rationally allocate medical and health hardware and software resources according to population, geography, demand, and other factors in order to improve efficiency. Accelerate the development of key health projects in Hainan Province. For example, promote the implementation of the global "One Health" demonstration project in Hainan Province through publicity and improvement of the medical and prevention integration governance model; Through the high-level equipment of hardware and software facilities, increase the degree of policy openness, pay attention to medical innovation and achievement transformation, and promote the process of the Boao Lecheng International Medical Tourism Pilot Zone; Based on the unique natural environment of Hainan Province, it is necessary to promote the rational development of the healthcare industry. 

Second, Hainan should balance the layout and optimize the allocation of health resources, and speed up the process of promoting graded diagnosis and treatment and "Internet Plus" medical treatment. Strengthen supervision and management, appropriately expand the scale of medical services in big cities, and for inefficient cities and counties, and improve the efficiency of service utilization by means of technology introduction, graded diagnosis and treatment, and "Internet +" medical treatment while ensuring adequate input. Speed up the building of economic circles in Haikou, Sanya, and Dan Ocean, form a point with a circle and a circle with an area, promote the formation of a new development pattern, accelerate the implementation of the relevant implementation opinions of Hainan Province's grid-tight medical and health service system, improve the medical consortium, hospital group, and other division of labor and cooperation models, and comprehensively improve diagnosis and treatment efficiency and resource utilization efficiency. 

Third, accelerate the introduction and training of health professionals, expand the total number, and improve the quality. On the one hand, the establishment of first-class medical colleges, especially nursing colleges. Independently cultivate and stabilize health professionals. On the other hand, Hainan Province can rely on favorable policies to improve the salary of health professionals, innovate the post and staffing system, focus on introducing and training high-level health professionals, master the core medical technology, improve the level of medical technology, and create high-end medical services in the world. In addition, the policy should be appropriately tilted to the grass-roots level, accelerate the construction of the grass-roots health personnel team, innovate the assessment system of grass-roots health personnel, pay more attention to the improvement of their post-competence, service attitude, diagnosis and treatment level, and ensure the quality and quantity of grass-roots health personnel. 

Fourth, strengthen international cooperation and exchanges to promote the sharing of medical and health resources. To maintain close contact with Beijing, Shanghai, Guangdong Province, and other places to strengthen technical exchanges and improve the level of medical technology and service efficiency. The introduction of domestic and foreign first-class medical institutions and research centers in Hainan to achieve co-construction, co-governance, and sharing to serve the construction of the Hainan Free Trade Port.

Other modifications:

We are very sorry. We examined the manuscript carefully and found that there were some sentences that needed to be changed and some places that had been added. These are highlighted in red in the manuscript. The details are as follows:

1.Line 20

The modified：Further description of co-contributing authors.

2.Lines 98-99

Original text：Hainan Province is the only free trade port in China

The modified：Hainan Province is currently the only free trade port with Chinese characteristics on the Chinese mainland,

3.Lines 140-141：Delete: Data envelopment analysis

Original text：Data envelopment analysis (DEA) is a non-parametric efficiency evaluation method first proposed by A. Charnes et al. in 1978.[9]

The modified：DEA is a non-parametric efficiency evaluation method first proposed by A. Charnes et al. in 1978.[13]

4.Lines 138, 206-207, 223, 235-236, 245-246

The modified：Bold font for the titles of the figures and tables

5.Lines 222：Delete Figure 1 and upload it separately.

6.Lines 301, 302, 305-306

Replace "Conclusion and policy implications" with "Conclusions and Suggestions" (Line 301)

Added content: The study found that (Line 302)

Added content: The, in Hainan Province (Lines 305-306)

7.Lines 308-312

Original text：In order to promote the high-quality development of the health industry of the Hainan Free Trade Port, the following suggestions are put forward. 

The modified：Improving the efficiency of the allocation of medical and health resources is an important guarantee for promoting the development of the health industry of the Hainan Free Trade Port and an inevitable requirement for the high-quality construction of the Hainan Free Trade Port. In order to improve the efficiency of medical and health resources in Hainan Free Trade Port, the following suggestions are put forward. 

8.As the "Introduction" section has been added, references have been added accordingly. (Lines 397-398, 403-412). The details are as follows: (With the revised manuscript, marked in yellow)

3. Chen Y. Research on equity and efficiency of health resource allocation in Yunnan Province [D]. Kunming University of Science and Technology,2021. 

5. Lacko R, Hajduová Z, Bakalár T, Pavolová H. Efficiency and Productivity Differences in Healthcare Systems: The Case of the European Union. Int J Environ Res Public Health. 2022 Dec 22;20(1):178. doi: 10.3390/ijerph20010178. PMID: 36612499; PMCID: PMC9819540.

6. Xiang Yingru, WU Jin, WANG Cong, HUA Le, HUANG Xianhong. Evaluation of the allocation efficiency of provincial health resources before and after medical reform: Based on DEA and Tobit Methods [J]. Soft Sciences of Health, 2019,34(09):71-75.

7. Luo Liangqing, Hu Meiling. An Analysis on the Productive Efficiency of China's Regional Health and Care Services [J]. Journal of Statistics and Information,2008,No.89(02):47-51.

9.Modify the year and page number of a reference. (Lines 434-435)

Original text：10. Chiu Y, Chen Y. The analysis of Taiwanese bank efficiency: incorporating both external environment risk and internal risk. Econ Model. 2017; 26:456–63. 

The modified：14. Chiu Y, Chen Y. The analysis of Taiwanese bank efficiency: incorporating both external environment risk and internal risk. Econ Model. 2009; 26:456–463. 

Thank you very much for your attention and time. Looking forward to your reply.

Yours sincerely,

Yanhua GONG

---

## [Decision Letter · Decision Letter 1]

30 Aug 2023

PONE-D-23-10563R1Study on the allocation efficiency of medical and health resources in Hainan Province under the construction of Free Trade Port: Based on the super-efficiency SBM — Malmquist modelPLOS ONE

Dear Dr. Gong,

Thank you for submitting your manuscript to PLOS ONE. After careful consideration, we feel that it has merit but does not fully meet PLOS ONE’s publication criteria as it currently stands. Therefore, we invite you to submit a revised version of the manuscript that addresses the points raised during the review process.

I would like to thank all Authors for the revised version of the manuscript. In my opinion, as confirmed by Reviewers, changes contributed to improved quality of the paper. However, there are still three points, I need to address.While you have thoroughly addressed Reviewers' comments in your reply, in my opinion some of those explanations need to be included in the main body of the paper in order to make your approach understandable for future readers. For example, you have explained in the reply what you understand under "institutions" (referring to an input variable), but it is still not clear for a Reader who reads the manuscript only. Since "institutions" may also stand for the social structures (which is a popular concept in the economics), it is needed for you to disambiguate. It would be also very beneficial to include your answer for 5th comment of the Reviewer #1 ("What evidence or examples are provided to support the claim that economically developed areas with large populations have relatively higher allocation efficiency of health resources in Hainan Province?") as well as 2nd and 3rd comment from Rewiever #2 into the main text ("It is incomprehensible to refer to a free port, what impact does it have on the functioning of health care? Are there any dependencies?", "The authors focus on human resources, and is it not worth considering other resources, there is also a budget constraint that affects the efficiency of management, and thus also clinical effectiveness").I'd also suggest considering removal of the "free-port" from the title, as suggested by Reviewer #3 ("(1) I do not suggest the authors to take the construction of free trade port in the title, because this point was not be discussed carefully. If possible, a DID or some other empirical estimation could be performed to estimate the policy effect, e.g. the construction of FTP."), since you do not in fact discuss the free-port as a determinant, more as a background factor. However, following this advice is not a necessary condition for the acceptance of the paper - the current title summarizes the study well.Rewiever #3 suggests some further polishing of the language and I agree with them. Some sentences could be improved. Proof-reading by a naitve speaker may be helpful in this area.

I do believe, that above-listed changes will be sufficient. 

We look forward to receiving your revised manuscript.

Kind regards,

Agata Sielska, Ph.D.

Academic Editor

PLOS ONE

Journal Requirements:

Reviewers' comments:

Reviewer's Responses to Questions

**Comments to the Author**

1. If the authors have adequately addressed your comments raised in a previous round of review and you feel that this manuscript is now acceptable for publication, you may indicate that here to bypass the “Comments to the Author” section, enter your conflict of interest statement in the “Confidential to Editor” section, and submit your "Accept" recommendation.

Reviewer #2: All comments have been addressed

Reviewer #3: (No Response)

Reviewer #4: All comments have been addressed

2. Is the manuscript technically sound, and do the data support the conclusions?

Reviewer #2: Yes

Reviewer #3: Partly

Reviewer #4: Yes

3. Has the statistical analysis been performed appropriately and rigorously? 

Reviewer #2: Yes

Reviewer #3: Yes

Reviewer #4: N/A

4. Have the authors made all data underlying the findings in their manuscript fully available?

Reviewer #2: Yes

Reviewer #3: Yes

Reviewer #4: No

5. Is the manuscript presented in an intelligible fashion and written in standard English?

Reviewer #2: Yes

Reviewer #3: No

Reviewer #4: Yes

6. Review Comments to the Author

Reviewer #2: The authors treated seriously the comments and requests for supplementation that I included in the review. They supplemented the article with missing - in my opinion - elements, as well as extended and clarified comments and conclusions as well as suggestions for decision-makers. They actually did it exhaustively. Therefore, I believe that the revised version of the article can be published.

Reviewer #3: This paper uses SBM-Malmquist model to estimate the allocative efficiency of healthcare resources in Hainan, China with county-level data. Although it may bring some information for us to understand the efficiency change for China's healthcare resources utilization, some major problems should be responded as the following.

(1) I do not suggest the authors to take the construction of free trade port in the title, because this point was not be discussed carefully. If possible, a DID or some other empirical estimation could be performed to estimate the policy effect, e.g. the construction of FTP.

(2) The main contribution and research questions was not reported carefully. The Introduction section should be rewritten and include a literature review to link the RQ, and the main contribution. "Foreign studies" should not be used.

(3) The methodology should be reported according to the marginal contribution of this manuscript, where "other indicators" is very clear. Maybe, the usage of per capita value is better.

(4)If possible, longer study period could contribute more.

(5) The discussion should be reorganized according to the main findings, contributions, and a detailed discussion for related literature and study for other provinces of China is needed.

(6) The last section is very lengthy and most of the suggestions are unnecessary without solid research findings. A more concise conclusion and policy implication section could bring more flowability.

(6) The language level should be improved, and the authors should report the work to international readers with its marginal contribution.

Reviewer #4: Deare authors, thank you for your revised version and I have read your manuscript. I have no other questions.

7. PLOS authors have the option to publish the peer review history of their article (what does this mean?). If published, this will include your full peer review and any attached files.

Reviewer #2: No

Reviewer #3: No

Reviewer #4: No

---

## [Author Response · Author response to Decision Letter 1]

22 Sep 2023

Dear Editor and Reviewers:

Thank you very much for your comments and professional advice on the manuscript (Title: Study on the allocation efficiency of medical and health resources in Hainan Province: Based on the super-efficiency SBM — Malmquist model. Number: PONE-D-23-10563R1). These opinions are helpful to improve the academic and rigorous nature of the manuscript. In the last round of revision, we carefully revised the manuscript and answered all the questions of the reviewers. Here we would like to thank the reviewers of the last round for their professional advice and recognition of our revised manuscript. Similarly, this time we are also holding a serious and steadfast learning attitude. According to your suggestions, we have carefully revised the manuscript and adjusted and polished some of the language. We hope the manuscript can be better. We have not changed much in the overall framework, but we changed the title to "Study on the allocation efficiency of medical and health resources in Hainan Province: Based on the super-efficiency SBM — Malmquist model" at your suggestion. In addition, the "Introduction" section and the "Discussion" section have been greatly revised. And the "Conclusions and suggestions" section has been deleted and reorganized. These are the focus of attention of experts. In a marked revision manuscript, we have highlighted your proposed revision in yellow, where the red font is to highlight that the author has added the editor's professional advice. Also, in this reply, we have highlighted some of the content in gray so that you can look more clear. Specific modification details are as follows:

Dr. Agata Sielska (Editor) :

Thank you very much for your professional advice on our first round of revisions. These suggestions help us to improve manuscript quality, make it more in line with journal requirements and attract readers' interest. We have added your suggestions to the body of the article. The following are our amendments and replies.

1.You have explained in the reply what you understand under "institutions" (referring to an input variable), but it is still not clear for a Reader who reads the manuscript only. Since "institutions" may also stand for the social structures (which is a popular concept in the economics), it is needed for you to disambiguate.

The author's answer：Thank you very much for your suggestion. We have added this to the article under Table 1 in the "Selection of indicators" section. (Lines 137-140) Specifically add as follows:

Added content: a Institutions mainly include hospitals, primary medical and health institutions, professional public health institutions, internet hospitals, and other medical and health institutions in the Hainan Provincial Health Statistics Yearbook. It reflects the overall size of the region's health systems. It is a public place to provide medical and health services and is one of the carriers.

2.It would be also very beneficial to include your answer to 5th comment from the Reviewer #1 in the main text ("What evidence or examples are provided to support the claim that economically developed areas with large populations have relatively higher allocation efficiency of health resources in Hainan Province?")

The author's answer：Thank you very much for your suggestion. We have already added this content to the "Discussion" section of the article at the appropriate place. At the same time, we also cite some studies by other scholars to illustrate this point. (Lines 298-320) Specifically add as follows:

Added content: The obvious differences among regions also reflect the unbalanced efficiency of medical resource allocation. The economically developed and populous region of Hainan Province has relatively high health resource allocation efficiency. Haikou City is the capital city of Hainan Province, and its annual gross domestic product (GDP) and population are the largest. Table 2 from the study also found that Haikou City has the highest overall efficiency. This may be related to the government's deepening reform of the healthcare system, adequate investment, and management. Economically developed regions usually have higher levels of investment in health resources, including medical facilities, equipment, and personnel, and such investment can provide better allocation and utilization of health resources and improve efficiency. They usually have more advanced medical technologies and management experience, which can help optimize health resource allocation and improve efficiency. For example, the application of 5G technology and electronic medical records can facilitate hierarchical treatment and improve the efficiency of healthcare organizations. The research and development of new drugs and improvements in surgical techniques can improve the efficiency of health resource allocation. Developed regions usually have better health policies and can receive greater policy concessions. For example, Haikou has many targeted health system integration and innovation measures. Many pilot measures will be implemented first in Haikou City. Areas with large populations usually have higher demand for health services, which prompts the government and healthcare organizations to increase investment in health. In addition, economically developed and populous areas are likely to be better educated and more aware of health care, which also drives the efficient use of health resources. This is consistent with Hang Ranfeng's study of health resource allocation efficiency in Jiangsu Province (2022) [18] and Huang Lili's study of health resource allocation efficiency in Hunan Province (2022) [19] .

3.It would be also very beneficial to include your answer to 2nd comment from Rewiever #2 in the main text ("It is incomprehensible to refer to a free port, what impact does it have on the functioning of health care? Are there any dependencies?"

The author's answer：Thank you very much for your suggestion. We have already added this content to the "Introduction" section of the article at the appropriate place. (Lines 91-101) Specifically add as follows:

Added content: The problem of unbalanced regional development has always existed in China. Hainan Province is currently the only free trade port with Chinese characteristics in mainland China, which is also known as the "test bed of national reform and opening up", which means that the successful experience of Hainan's reforms can be extended to other regions of China, and its construction is of great significance to China. [6] In the context of the free trade port, Hainan will have more development opportunities. At the same time, the efficiency of healthcare resource allocation, as an important indicator for safeguarding people's health, has also received widespread attention. Hainan's free trade port policy plays an important role in the operation of healthcare, for example, by providing more medical resources, promoting medical technology innovation, improving the quality of medical services, promoting the development of medical tourism, and optimizing the medical management system.

4.It would be also very beneficial to include your answer to 2nd comment from Rewiever #2 in the main text ("The authors focus on human resources, and is it not worth considering other resources, there is also a budget constraint that affects the efficiency of management, and thus also clinical effectiveness").

The author's answer：Thank you very much for your suggestion. We have already added this content to the "Discussion" section of the article at the appropriate place. (Lines 278-296) Specifically add as follows:

Added content: The reason for this is that, on the one hand, Hainan Province is unable to improve healthcare output through scientific and technological progress and innovation. On the other hand, the current investment in healthcare by the government of Hainan Province is too focused on hardware elements, and to a certain extent, it has neglected the improvement of healthcare technology, the quality of healthcare services, and the quality of health personnel. Of course, hardware inputs are essential, especially financial inputs. But these human resources inputs in health care are also very important. Medical and healthcare human resources are the core of the development of healthcare, which is a key factor. Human resources for health care have a significant impact on the development of health care in Hainan's free trade port and are important for the provision of high-quality health services, the promotion of medical tourism, and the protection of health security. Adequate human resources for health can provide a wider range and quality of health services and ensure that residents have timely access to medical and healthcare services. The development of a free trade port in Hainan implies the possibility of population mobility and growth, thus requiring sufficient human resources for health to meet the growing demand for health services. As a province with a highly developed tourism industry, Hainan has sufficient human resources for health to provide a high level of medical services and attract patients to the province. At the same time, Hainan's free trade port will require stronger human resources to ensure health safety and epidemic prevention and control.

5.I'd also suggest considering removal of the "free-port" from the title, as suggested by Reviewer #3, since you do not in fact discuss the free-port as a determinant, more as a background factor. However, following this advice is not a necessary condition for the acceptance of the paper - the current title summarizes the study well.

The author's answer：Thank you very much for your professional advice. After consultation and thinking, we also agree to delete the title "under the construction of free trade port" and change it to "Study on the allocation efficiency of medical and health resources in Hainan Province: Based on the super-efficiency SBM — Malmquist model". But the free trade port is still a background content of this study, in this background to explore a new development of Hainan Province. The free trade port policy has given Hainan a new opportunity for development. (Lines4-5)

Original text：Study on the allocation efficiency of medical and health resources in Hainan Province under the construction of Free Trade Port: Based on the super-efficiency SBM — Malmquist model

The modified：Study on the allocation efficiency of medical and health resources in Hainan Province: Based on the super-efficiency SBM — Malmquist model

6.Rewiever #3 suggests some further polishing of the language and I agree with them. Some sentences could be improved. Proof-reading by a naitve speaker may be helpful in this area.

The author's answer：Thank you very much for your suggestions. In order to improve our manuscript language, we consulted a specialized polishing agency before submitting the manuscript for the first time. After this revision, we discussed and revised the writing language seriously with a professional English tutor. All the suggestions he gave us were taken on board and adjusted accordingly in the manuscript. We apologize for causing you unnecessary trouble and hope that the revised version will meet your expectations.

(1) Lines 49-52

Original text：According to the "Healthy China 2030" Plan's Outline in 2016, by 2030, China's health institutions and systems will be improved, the development of the health sector will be more coordinated, healthy lifestyles will be popularized, and the quality and guarantee of health services will continue to improve.

The modified：The 2016 Outline of the "Healthy China 2030" Plan proposes that by 2030, China's health institution system will be more complete, the development of health undertakings will be more coordinated, healthy lifestyles will be popularized, and the quality of health services and the level of protection will be continuously improved. (Lines 49-52)

(2)Lines 244-246

Original text：However, through the super-efficiency SBM measurement, it is found that the comprehensive efficiency score of the samples is 0.975, indicating that the allocation efficiency of medical and health resources in Hainan Province is not well, which needs to be further improved. 

The modified：However, through the super-efficiency SBM measurement, it is found that the comprehensive efficiency score of the samples is 0.975, which indicates that the efficiency of healthcare resource allocation in Hainan Province is not high and needs to be further improved.

(3)Line 248

Original text：The main reasons for the decline may have been these reasons.

The modified：The main reasons for the decline may be these:

(4)Lines 278-283

Original text：On the one hand, Hainan Province is not able to improve its medical and health output through scientific and technological progress and innovation. On the other hand, the current investment of the Hainan Provincial government in the medical and health field focuses too much on the hardware elements and neglects to some extent the improvement of medical and health technology, medical service quality, and health personnel quality.

The modified：The reason for this is that, on the one hand, Hainan Province is unable to improve healthcare output through scientific and technological progress and innovation. On the other hand, the current investment in healthcare by the government of Hainan Province is too focused on hardware elements, and to a certain extent, it has neglected the improvement of healthcare technology, the quality of healthcare services, and the quality of health personnel. 

(5)Lines 341-346

Original text：The Hainan provincial government should further implement and optimize policies, accelerate the process of medical reform, improve detailed medical and health resource planning, and rationally allocate medical and health hardware and software resources according to population, geography, demand, and other factors in order to improve efficiency.

The modified：The Hainan government should further implement and optimize its policies, accelerate the process of healthcare reform, improve detailed healthcare resource planning, rationally allocate healthcare hardware and software resources based on demographic, geographic, and demand factors, and promote the expansion and balanced allocation of high-quality healthcare resources to improve efficiency.

Reviewer #3:

Thank you very much for your recognition of the manuscript and your professional suggestions. The following are our amendments and replies.

1.I do not suggest the authors to take the construction of free trade port in the title, because this point was not be discussed carefully. If possible, a DID or some other empirical estimation could be performed to estimate the policy effect, e.g. the construction of FTP.

The author's answer：Thank you very much for your professional advice. After consultation and thinking, we also agree to delete the title "under the construction of free trade port" and change it to "Study on the allocation efficiency of medical and health resources in Hainan Province: Based on the super-efficiency SBM — Malmquist model". But the free trade port is still a background content of this study, in this background to explore a new development of Hainan Province. The free trade port policy has given Hainan a new opportunity for development. (Lines4-5)

Original text：Study on the allocation efficiency of medical and health resources in Hainan Province under the construction of Free Trade Port: Based on the super-efficiency SBM — Malmquist model

The modified：Study on the allocation efficiency of medical and health resources in Hainan Province: Based on the super-efficiency SBM — Malmquist model

2.The main contribution and research questions was not reported carefully. The Introduction section should be rewritten and include a literature review to link the RQ, and the main contribution. "Foreign studies" should not be used.

The author's answer：Thank you very much for your professional advice. We have revised and improved the "Introduction" section. As it reflects the situation in China, we agree with your suggestion of not using some foreign studies. We have more prominently highlighted the major domestic studies and contributions in this area. We focus on reflecting the current reality in China as well as citing studies in other parts of China in this area, and we summarize the main contributions and shortcomings of the relevant literature in this area so as to better draw out the research questions of this paper. This study focuses on Hainan Province and is a study of how to improve the efficiency of healthcare resource allocation in Hainan Province. In the context of free trade port, this study adopts the super-efficient SBM model and Malmquist index to study the efficiency of healthcare resource allocation in Hainan Province from 2016 to 2020, analyze the status quo, problems, and reasons, and put forward rationalization suggestions to provide reference for the scientific formulation of healthcare development planning. (Lines 45-121)

Original text：The contradiction between the finiteness of health resources and the infinite demand of people is prominent.[1] How to make more effective use of the limited health resources, meet the needs of people at different levels of health services, optimize the distribution of health resources and boost the efficiency of use is a common problem faced by the global medical and health reform.[1, 2] According to the "Healthy China 2030" Plan's Outline in 2016, by 2030, China's health institutions and systems will be improved, the development of the health sector will be more coordinated, healthy lifestyles will be popularized, and the quality and guarantee of health services will continue to improve. The important premise of improving health levels and promoting health equity is the rational allocation and efficient utilization of medical and health resources. The improvement of the efficiency of the allocation of medical and health resources can not only promote the accessibility and quality of medical services but also play an important role in improving people's living standards and promoting economic growth. Since the implementation of China's new medical reform in 2009, the amount of health resource investment has generally increased, and great results have been achieved.

Foreign studies on the allocation of health resources started earlier, and the UK was the first country to conduct research on the allocation of health resources [3]. In 1976, the Resource Allocation Working Party (RAWP) was established in the United Kingdom to study the allocation of health resources. Research on the allocation of health resources in China began in the 1980s. With the development of the economy, the rapid increase in population, and the intensification of aging, people have put forward higher requirements for medical and health resources and services, and the government has continuously increased its investment in health resources. The number of health institutions, beds, and health technicians is increasing, and the service capacity of medical and health institutions is greatly improved, but the efficiency of medical and health resource allocation is still low, which may lead to increased investment in subsequent health resources but poor output. This not only fails to meet people's growing medical and health needs but also causes serious waste of medical and health resources, resulting in the rapid growth of medical costs, increasing people's medical burden, and restricting the sustainable and high-quality development of health and health undertakings, which is contrary to the ultimate goal of national health and health development.

How to make the best use of health resources is a hot topic for scholars at home and abroad. Data envelopment analysis ( DEA ) is the most mature and extensive method to study the efficiency of the allocation of medical and health resources. For example, Sayem Ahmed (2019) [4] used the DEA, Tobit model, and smooth bootstrap model to evaluate the technical efficiency of health systems in Asian countries and found that up to 91.3% of countries had inefficient utilization of health resources. Lacko R. (2022) [5] uses the DEA model and Malmquist index to compare the efficiency and productivity of EU countries. It is found that although countries joining the EU get a high technical efficiency value, the overall technical efficiency shows a downward trend, and the productivity of Western countries shows stagnation. Xiang Yingru (2020) [6] analyzed the efficiency of provincial health resources in China by using the BCC model of DEA and the Malmquist index. The study pointed out that after the implementation of medical reform in China, resources are still wasted, output is insufficient in some areas, and efficiency is low due to the excessive number of public hospitals and the serious aging of the population. Technological progress is a key factor affecting efficiency. Luo Liangqing (2008) [7] used the DEA model to study the production efficiency of medical and health services in the eastern, central, and western regions of China and found that the production efficiency of medical and health services in China was generally low. It can be seen from many related studies that the traditional DEA model has some limitations when studying the efficiency of the allocation of medical and health resources. Therefore, the method is improved in this article.

The problem of uneven development among different regions in China has always existed. Hainan Province is currently the only free trade port with Chinese characteristics on the Chinese mainland, and is also known as a "test bed for national reform and opening up ", which means that the successful experience of Hainan's reforms may be extended to other parts of China, and its construction is of great significance to China.[8] Hainan Province (108°37 '-- 111°03' east longitude, 18°10 '-- 20°10' north latitude) is located in the southernmost part of China, close to Southeast Asia, with a total land area of 35,400 square kilometers, and the number of permanent residents in 2020 is 10,123,400 people. In 2018, the Party Central Committee decided to support Hainan in gradually exploring and steadily deepening the construction of a free trade port with Chinese characteristics, which has helped the province comprehensively deepen reform. Hainan is committed to building Health Island and Longevity Island. Under the free trade port policy, Hainan has focused on institutional integration innovation and made major breakthroughs in health care. However, with the increase in population, the intensification of aging, and the change of disease spectrum, the current medical and health resources in Hainan Province are obviously in shortage, and there are outstanding problems such as imbalance, inequity, low efficiency, and lack of human resources. In order to effectively cope with future changes, Hainan must speed up the adjustment and optimization of medical and health resources and improve the efficiency of resource utilization, so as to provide an important guarantee for the high-quality development of health services in Hainan and escort the construction of free trade ports. At present, more attention is paid to the ecological environment and trade of Hainan Province at home and abroad, while there are few related types of research on health status, especially those on Hainan Province under the policy of free trade port with Chinese characteristics. In this study, the super-efficiency SBM model and Malmquist index were used to study the allocation efficiency of medical and health resources in Hainan Province during 2016-2020, analyze the current situation, problems, and causes, and put forward reasonable suggestions to provide a reference for scientific health development planning.

The modified：The contradiction between the limited nature of health resources and people's unlimited needs is prominent. [1] How to make more effective use of limited health resources, meet the needs of people at different levels for health services, optimize the allocation of health resources, and improve the efficiency of their use is a common problem facing global healthcare reform. [1,2] The 2016 Outline of the "Healthy China 2030" Plan proposes that by 2030, China's health institution system will be more complete, the development of health undertakings will be more coordinated, healthy lifestyles will be popularized, and the quality of health services and the level of protection will be continuously improved. An important prerequisite for improving health and promoting health equity is the rational allocation and effective utilization of medical and health resources. Improving the efficiency of healthcare resource allocation not only promotes the accessibility and quality of healthcare services but also plays an important role in raising people's living standards and promoting economic growth. Since the implementation of China's new healthcare reform in 2009, the investment in health resources has generally increased and achieved great results.

Research on health resource allocation in China began in the 1980s. With the development of the economy, the rapid increase in population, and the intensification of aging, people have put forward higher requirements for medical and health resources and services, and the government has continuously increased its investment in health resources. The number of health institutions, beds, and health technicians has continued to increase, and the service capacity of health institutions has greatly improved, but the efficiency of the allocation of health resources is still low, which may lead to subsequent increases in the input of health resources and poor output. This not only fails to meet the growing health care needs of the people but also results in a serious waste of health care resources, leading to a rapid increase in health care costs, aggravating the health care burden of the people, restricting the sustained and high-quality development of health care, and running counter to the ultimate goal of the country's health and health development. Ye Yizhong (2022) [19] used the three-stage SBM model and Malmquist index method to statically and dynamically evaluate the allocation efficiency of healthcare resources in 31 provinces (autonomous regions and municipalities directly under the central government, except for Hong Kong, Macao, and Taiwan) in China from 2015 to 2020. His study found that the allocation efficiency of health resources in China in 2015–2020 is low and on a downward trend, with obvious constraints on technological progress, large differences in allocation efficiency between regions and provinces, and significant impacts on the gross domestic product of each region, the number of permanent residents, and the level of urbanization. Xiang Yingru (2020) [6] analyzed the efficiency of health resources at the provincial level in China using the BCC model of DEA and the Malmquist index. The study pointed out that after the implementation of China's health care reform, due to the excessive number of public hospitals and the serious aging of the population, there are still problems such as waste of resources, insufficient output in some areas, and inefficiency. Technological progress is a key factor affecting efficiency. Luo Liangqing (2008) [7] utilized the DEA model to study the production efficiency of healthcare services in China's eastern, central, and western regions and found that China's healthcare service production efficiency is generally low. A large number of research results provide theoretical guidance for the research idea in this paper. From many related studies, it can be concluded that DEA is the most mature and widely used method to study the efficiency of healthcare resource allocation. However, the traditional DEA model has certain limitations in studying healthcare resource allocation efficiency. Therefore, this paper improves the method.

The problem of unbalanced regional development has always existed in China. Hainan Province is currently the only free trade port with Chinese characteristics in mainland China, which is also known as the "test bed of national reform and opening up", which means that the successful experience of Hainan's reforms can be extended to other regions of China, and its construction is of great significance to China. [6] In the context of the free trade port, Hainan will have more development opportunities. At the same time, the efficiency of healthcare resource allocation, as an important indicator for safeguarding people's health, has also received widespread attention. Hainan's free trade port policy plays an important role in the operation of healthcare, for example, by providing more medical resources, promoting medical technology innovation, improving the quality of medical services, promoting the development of medical tourism, and optimizing the medical management system. Hainan Province (108°37′–111°03′ E, 18°10′–20°10′ N) is located in the southernmost part of China, close to Southeast Asia, with a land area of 35,400 square kilometers and a resident population of 10,123,400 in 2020. In 2018, the Central Committee of the CPC decided to support Hainan to progressively explore and steadily deepen the construction of a free trade port with Chinese characteristics, helping Hainan to deepen its reforms comprehensively. Hainan is committed to building a healthy and long-lived island. Under the free trade port policy, Hainan focuses on system integration and innovation and has made significant breakthroughs in the field of healthcare. with the increase in population, the intensification of aging, and the change of disease spectrum, the current allocation of healthcare resources in Hainan Province is unreasonable, and the quality of healthcare services and the utilization of healthcare resources are inefficient, which seriously restricts the sustainable development of healthcare and wellness. Therefore, it is of great significance for Hainan Province to study how to improve the efficiency of healthcare resource allocation in Hainan Province to meet the people's demand for healthcare services, provide an important guarantee for the high-quality development of Hainan's healthcare industry, and escort the construction of the Free Trade Port. In the context of free trade port, this study adopts the super-efficient SBM model and Malmquist index to study the efficiency of healthcare resource allocation in Hainan Province from 2016 to 2020, analyze the status quo, problems, and reasons, and put forward rationalization suggestions to provide reference for the scientific formulation of healthcare development planning.

3.The methodology should be reported according to the marginal contribution of this manuscript, where "other indicators" is very clear. Maybe, the usage of per capita value is better. 

The author's answer：Thank you very much for your suggestions. This paper has mentioned some of the marginal contributions that can be reacted to by this study in the “Discussion” section. For example, when analyzing the reasons for the decrease in comprehensive efficiency, the second reason is mentioned that Hainan Province has a corresponding increase in capital, equipment, and talent under the policy of free trade port, but the efficiency decreases instead of increasing. (Lines 258-267) When analyzing the results of dynamic efficiency, it is also found that technical efficiency, pure technical efficiency, and scale efficiency were all positive in 2016-2017, indicating that technical efficiency contributes to the increase of total factor productivity. The decline of total factor productivity in 2017 - 2020 is affected by both scale efficiency and technological progress, and it is more affected by technological setbacks. (Lines 273-277) When analyzing regional efficiency differences, this study also mentions that the management model should be reformed and medical services optimized to improve utilization efficiency, rather than compressing inputs, which also indicates that over time and with policy changes, these need to be innovated and injected with "fresh blood". (Lines 324-326) The application of marginal effects We focus on exploring the extent to which input influences have a marginal impact on the allocation efficiency of medical and health resources. (This is what we are currently working on.)

Using value per capita (which we call gross domestic product per capita) as an indicator is a very good suggestion. The "Other indicators" are our auxiliary indicators. We are thinking more of the gross domestic product per capita (GDP per capita) as our next study of impact factors. This we think will be more meaningful. The level of social and economic development directly affects government finance and residents' living standards, affects the accessibility of medical and health services, and further affects the supply and demand of medical and health resources. The important influence of the level of economic development on the allocation efficiency of healthcare resources can also be fully realized from the study of Wei Kezhen, Zhou Liting, and others.

[1] Wei Kezhen. Research on measurement and influencing factors of allocation efficiency of medical and health resources in China [D]. Shaanxi Normal University,2018.

[2] Zhou L T. Study on efficiency of medical and health resource allocation and its influencing factors in Jiangsu Province [D]. Harbin University of Commerce,2022.

Article Related Content:

Second, it may have something to do with the construction of free trade ports in Hainan Province starting in 2018. Hainan has given priority to institutional integration and innovation in its development. Various preferential policies have been formulated and introduced to attract capital, talent, high-end equipment, and other resources to Hainan, breaking the state of self-sufficiency in the past. With the deepening of the reform of the medical system, medical and health resources are being restructured and adjusted, and there is a phenomenon of underutilization and reasonable operation, resulting in a decline in efficiency instead of improvement. This also reflects from another side that no matter how much health resources are invested, if the resources are not reasonably and fully utilized, the efficiency of resource allocation will decline and the output will be poor. (Lines 258-267) 

Malmquist index shows that technical efficiency, pure technical efficiency, and scale efficiency have positive growth in 2016-2017, indicating that technical efficiency contributes to the improvement of total factor productivity. The decline of total factor productivity from 2017 to 2020 is influenced by both scale efficiency and technological progress and is greatly influenced by technological regress. (Lines 273-277) 

Therefore, it is necessary for each district and county government to reform the management mode, optimize the medical service and improve the service utilization efficiency, rather than compress the input. (Lines 324-326) 

4.If possible, longer study period could contribute more.

The author's answer：We started this research work in August 2021. After repeated research, searching for information, consulting relevant experts, and continuous revision, we have enough data and evidence to do this research. During the period, we wrote the "High-quality Development Report of Hainan Health Industry in 2021" and "High-quality Development Report of Hainan Health Industry in 2022" for the Hainan Provincial government, both of which were highly evaluated by them and considered to be operable and practical. In addition, we have also published articles on the allocation of medical and health resources in Hainan Province in relevant journals at home and abroad. We are well aware of the importance of the allocation of medical and health resources, and we also know that perhaps our current research is not enough, and there are still places to study and study hard. We will do our best to make greater contributions in this area.

5.The discussion should be reorganized according to the main findings, contributions, and a detailed discussion for related literature and study for other provinces of China is needed.

The author's answer：Thank you very much for your suggestions. We have reorganized and revised the "Discussion" section and introduced relevant studies from other provinces in China. The "Discussion" section includes the following points: the comprehensive situation, dynamic situation and regional differences of the allocation efficiency of medical and health resources in Hainan Province. Studies from other Chinese regions, such as Chongqing, Jiangsu Province, and Hunan Province, have also been introduced. (Lines 242-326) 

Original text：With the construction of the Hainan Free Trade Port, the demand for population and health services increases, and the allocation of health resources also increases year by year. However, through the super-efficiency SBM measurement, it is found that the comprehensive efficiency score of the samples is 0.975, indicating that the allocation efficiency of medical and health resources in Hainan Province is not well, which needs to be further improved. 2017 was relatively good, but the comprehensive efficiency from 2019 to 2020 was all less than 1, and there was a downward trend. The main reasons for the decline may have been these reasons. First, in 2017, it was emphasized in the Key Tasks of Deepening the Reform of the Medical and Health System in 2017 that the year 2017 was a key year for China to form a systematic framework of the basic medical and health system and complete the phased objectives of the medical reform. Therefore, China promulgated a number of medical reform policies, such as the cancellation of the drug price addition policy, the comprehensive launch of the medical consortium pilot work, and the implementation of a hierarchical diagnosis and treatment system. The medical and health service system in our country has broken the inherent mode under the effects of this series of medical reforms. During the initial period of the implementation of the new and old policies, it has exerted influence on the normal operation of health undertakings. Second, it may have something to do with the construction of free trade ports in Hainan Province starting in 2018. Hainan has given priority to institutional integration and innovation in its development. Various preferential policies have been formulated and introduced to attract capital, talent, high-end equipment, and other resources to Hainan, breaking the state of self-sufficiency in the past. With the deepening of the reform of the medical system, medical and health resources are being restructured and adjusted, and there is a phenomenon of underutilization and reasonable operation, resulting in a decline in efficiency instead of improvement. This also reflects from another side that no matter how much health resources are invested, if the resources are not reasonably and fully utilized, the efficiency of resource allocation will decline and the output will be poor. Third, due to the impact of the global COVID-19 epidemic in 2019–2020, health resources have been urgently deployed, including equipment, beds, personnel and other resources, focusing on the allocation of the most affected areas. More beds were transferred to treat COVID-19 patients. Medical resources are diverted to emergency response to the epidemic, thus affecting the original efficiency.

Malmquist index shows that technical efficiency, pure technical efficiency, and scale efficiency have positive growth in 2016-2017, indicating that technical efficiency contributes to the improvement of total factor productivity. The decline of total factor productivity from 2017 to 2020 is influenced by both scale efficiency and technological progress and is greatly influenced by technological regress. On the one hand, Hainan Province is not able to improve its medical and health output through scientific and technological progress and innovation. On the other hand, the current investment of the Hainan Provincial government in the medical and health field focuses too much on the hardware elements and neglects to some extent the improvement of medical and health technology, medical service quality, and health personnel quality.

The obvious differences among regions also reflect the unbalanced efficiency of medical resource allocation. From the perspective of comprehensive efficiency, Haikou, as the capital city of Hainan Province, has a relatively developed economy and a better allocation efficiency of health resources, which may be related to the government's deepening of the reform of the medical system, with sufficient input and in place management. However, from the perspective of total factor productivity, Haikou City is restricted by scale efficiency, indicating that the medical scale of Haikou City should be properly expanded. Inefficient cities and counties with large areas and small populations, there are redundant inputs and insufficient output, and it is difficult to make full use of health resources. Therefore, it is necessary for each district and county government to reform the management mode, optimize the medical service and improve the service utilization efficiency, rather than compress the input.

The modified：With the construction of the Hainan Free Trade Port, the demand for population and health services increases, and the allocation of health resources also increases year by year. However, through the super-efficiency SBM measurement, it is found that the comprehensive efficiency score of the samples is 0.975, which indicates that the efficiency of healthcare resource allocation in Hainan Province is not high and needs to be further improved. 2017 was relatively good, but the comprehensive efficiency from 2019 to 2020 was all less than 1, and there was a downward trend. The main reasons for the decline may be these: First, in 2017, it was emphasized in the Key Tasks of Deepening the Reform of the Medical and Health System in 2017 that the year 2017 was a key year for China to form a systematic framework of the basic medical and health system and complete the phased objectives of the medical reform. Therefore, China promulgated a number of medical reform policies, such as the cancellation of the drug price addition policy, the comprehensive launch of the medical consortium pilot work, and the implementation of a hierarchical diagnosis and treatment system. The medical and health service system in our country has broken the inherent mode under the effects of this series of medical reforms. During the initial period of the implementation of the new and old policies, it has exerted influence on the normal operation of health undertakings. Second, it may have something to do with the construction of free trade ports in Hainan Province starting in 2018. Hainan has given priority to institutional integration and innovation in its development. Various preferential policies have been formulated and introduced to attract capital, talent, high-end equipment, and other resources to Hainan, breaking the state of self-sufficiency in the past. With the deepening of the reform of the medical system, medical and health resources are being restructured and adjusted, and there is a phenomenon of underutilization and reasonable operation, resulting in a decline in efficiency instead of improvement. This also reflects from another side that no matter how much health resources are invested, if the resources are not reasonably and fully utilized, the efficiency of resource allocation will decline and the output will be poor. Third, due to the impact of the global COVID-19 epidemic in 2019–2020, health resources have been urgently deployed, including equipment, beds, personnel and other resources, focusing on the allocation of the most affected areas. More beds were transferred to treat COVID-19 patients. Medical resources are diverted to emergency response to the epidemic, thus affecting the original efficiency.

Malmquist index shows that technical efficiency, pure technical efficiency, and scale efficiency have positive growth in 2016-2017, indicating that technical efficiency contributes to the improvement of total factor productivity. The decline of total factor productivity from 2017 to 2020 is influenced by both scale efficiency and technological progress and is greatly influenced by technological regress. This is consistent with the studies of Cai Zihan (2022) [16] and Tian Haoguo (2022) [17]. The reason for this is that, on the one hand, Hainan Province is unable to improve healthcare output through scientific and technological progress and innovation. On the other hand, the current investment in healthcare by the government of Hainan Province is too focused on hardware elements, and to a certain extent, it has neglected the improvement of healthcare technology, the quality of healthcare services, and the quality of health personnel. Of course, hardware inputs are essential, especially financial inputs. But these human resources inputs in health care are also very important. Medical and healthcare human resources are the core of the development of healthcare, which is a key factor. Human resources for health care have a significant impact on the development of health care in Hainan's free trade port and are important for the provision of high-quality health services, the promotion of medical tourism, and the protection of health security. Adequate human resources for health can provide a wider range and quality of health services and ensure that residents have timely access to medical and healthcare services. The development of a free trade port in Hainan implies the possibility of population mobility and growth, thus requiring sufficient human resources for health to meet the growing demand for health services. As a province with a highly developed tourism industry, Hainan has sufficient human resources for health to provide a high level of medical services and attract patients to the province. At the same time, Hainan's free trade port will require stronger human resources to ensure health safety and epidemic prevention and control.

The obvious differences among regions also reflect the unbalanced efficiency of medical resource allocation. The economically developed and populous region of Hainan Province has relatively high health resource allocation efficiency. Haikou City is the capital city of Hainan Province, and its annual gross domestic product (GDP) and population are the largest. Table 2 from the study also found that Haikou City has the highest overall efficiency. This may be related to the government's deepening reform of the healthcare system, adequate investment, and management. Economically developed regions usually have higher levels of investment in health resources, including medical facilities, equipment, and personnel, and such investment can provide better allocation and utilization of health resources and improve efficiency. They usually have more advanced medical technologies and management experience, which can help optimize health resource allocation and improve efficiency. For example, the application of 5G technology and electronic medical records can facilitate hierarchical treatment and improve the efficiency of healthcare organizations. The research and development of new drugs and improvements in surgical techniques can improve the efficiency of health resource allocation. Developed regions usually have better health policies and can receive greater policy concessions. For example, Haikou has many targeted health system integration and innovation measures. Many pilot measures will be implemented first in Haikou City. Areas with large populations usually have higher demand for health services, which prompts the government and healthcare organizations to increase investment in health. In addition, economically developed and populous areas are likely to be better educated and more aware of health care, which also drives the efficient use of health resources. This is consistent with Hang Ranfeng's study of health resource allocation efficiency in Jiangsu Province (2022) [18] and Huang Lili's study of health resource allocation efficiency in Hunan Province (2022) [19] . However, from the perspective of total factor productivity, Haikou City is restricted by scale efficiency, indicating that the medical scale of Haikou City should be properly expanded. Inefficient cities and counties with large areas and small populations, there are redundant inputs and insufficient output, and it is difficult to make full use of health resources. Therefore, it is necessary for each district and county government to reform the management mode, optimize the medical service and improve the service utilization efficiency, rather than compress the input.

6.The last section is very lengthy and most of the suggestions are unnecessary without solid research findings. A more concise conclusion and policy implication section could bring more flowability.

The author's answer：After careful consideration and discussion, we agree with your views. We have made deletions and minor changes to the "suggestions" section to make the suggestions more flexible and operational. Specific modifications are as follows. (Lines 340-372) 

Original text：First, Hainan should continue to take institutional integration and innovation as the "starting point" to improve the ability of modern governance of health services. The "14th Five-Year Plan for Hygiene and Health of Hainan Province" issued by Hainan Province in 2021 emphasizes that Hainan must accelerate the expansion and balanced distribution of high-quality medical and health resources and improve the allocation efficiency of medical and health resources. The Hainan provincial government should further implement and optimize policies, accelerate the process of medical reform, improve detailed medical and health resource planning, and rationally allocate medical and health hardware and software resources according to population, geography, demand, and other factors in order to improve efficiency. Accelerate the development of key health projects in Hainan Province. For example, promote the implementation of the global "One Health" demonstration project in Hainan Province through publicity and improvement of the medical and prevention integration governance model; Through the high-level equipment of hardware and software facilities, increase the degree of policy openness, pay attention to medical innovation and achievement transformation, and promote the process of the Boao Lecheng International Medical Tourism Pilot Zone; Based on the unique natural environment of Hainan Province, it is necessary to promote the rational development of the healthcare industry. 

Second, Hainan should balance the layout and optimize the allocation of health resources, and speed up the process of promoting graded diagnosis and treatment and "Internet Plus" medical treatment. Strengthen supervision and management, appropriately expand the scale of medical services in big cities, and for inefficient cities and counties, and improve the efficiency of service utilization by means of technology introduction, graded diagnosis and treatment, and "Internet +" medical treatment while ensuring adequate input. Speed up the building of economic circles in Haikou, Sanya, and Dan Ocean, form a point with a circle and a circle with an area, promote the formation of a new development pattern, accelerate the implementation of the relevant implementation opinions of Hainan Province's grid-tight medical and health service system, improve the medical consortium, hospital group, and other division of labor and cooperation models, and comprehensively improve diagnosis and treatment efficiency and resource utilization efficiency. 

Third, accelerate the introduction and training of health professionals, expand the total number, and improve the quality. On the one hand, the establishment of first-class medical colleges, especially nursing colleges. Independently cultivate and stabilize health professionals. On the other hand, Hainan Province can rely on favorable policies to improve the salary of health professionals, innovate the post and staffing system, focus on introducing and training high-level health professionals, master the core medical technology, improve the level of medical technology, and create high-end medical services in the world. In addition, the policy should be appropriately tilted to the grass-roots level, accelerate the construction of the grass-roots health personnel team, innovate the assessment system of grass-roots health personnel, pay more attention to the improvement of their post-competence, service attitude, diagnosis and treatment level, and ensure the quality and quantity of grass-roots health personnel. 

Fourth, strengthen international cooperation and exchanges to promote the sharing of medical and health resources. To maintain close contact with Beijing, Shanghai, Guangdong Province, and other places to strengthen technical exchanges and improve the level of medical technology and service efficiency. The introduction of domestic and foreign first-class medical institutions and research centers in Hainan to achieve co-construction, co-governance, and sharing to serve the construction of the Hainan Free Trade Port.

The modified：First, Hainan should continue to take institutional integration and innovation as the "starting point" to improve the ability of modern governance of health services. The Hainan government should further implement and optimize its policies, accelerate the process of healthcare reform, improve detailed healthcare resource planning, rationally allocate healthcare hardware and software resources based on demographic, geographic, and demand factors, and promote the expansion and balanced allocation of high-quality healthcare resources to improve efficiency. Accelerate the development of key health projects in Hainan Province. 

Second, Hainan should balance the layout and optimize the allocation of health resources, and speed up the process of promoting graded diagnosis and treatment and "Internet Plus" medical treatment. Strengthen supervision and management, appropriately expand the scale of medical services in big cities, and for inefficient cities and counties, and improve the efficiency of service utilization by means of technology introduction, graded diagnosis and treatment, and "Internet +" medical treatment while ensuring adequate input. Accelerate the implementation of the relevant implementation opinions of Hainan Province's grid-tight medical and health service system, improve the medical consortium, hospital group, and other division of labor and cooperation models, and comprehensively improve diagnosis and treatment efficiency and resource utilization efficiency. 

Third, accelerate the introduction and training of health professionals, expand the total number, and improve the quality. On the one hand, the establishment of first-class medical colleges, especially nursing colleges. Independently cultivate and stabilize health professionals. On the other hand, Hainan Province can rely on favorable policies to improve the salary of health professionals, innovate the post and staffing system, focus on introducing and training high-level health professionals, master the core medical technology, improve the level of medical technology, and create high-end medical services in the world. In addition, the policy should be appropriately tilted to the grass-roots level and accelerate the construction of the grass-roots health personnel team. 

Fourth, strengthen international cooperation and exchanges to promote the sharing of medical and health resources. To maintain close contact with Beijing, Shanghai, Guangdong Province, and other places to strengthen technical exchanges and improve the level of medical technology and service efficiency. The introduction of domestic and foreign first-class medical institutions and research centers in Hainan to achieve co-construction, co-governance, and sharing to serve the construction of the Hainan Free Trade Port.

7.The language level should be improved, and the authors should report the work to international readers with its marginal contribution.

The author's answer：Thank you very much for your suggestions. In order to improve our manuscript language, we consulted a specialized polishing agency before submitting the manuscript for the first time. After this revision, we discussed and revised the writing language seriously with a professional English tutor. All the suggestions he gave us were taken on board and adjusted accordingly in the manuscript. We apologize for causing you unnecessary trouble and hope that the revised version will meet your expectations.

(1) Lines 49-52

Original text：According to the "Healthy China 2030" Plan's Outline in 2016, by 2030, China's health institutions and systems will be improved, the development of the health sector will be more coordinated, healthy lifestyles will be popularized, and the quality and guarantee of health services will continue to improve.

The modified：The 2016 Outline of the "Healthy China 2030" Plan proposes that by 2030, China's health institution system will be more complete, the development of health undertakings will be more coordinated, healthy lifestyles will be popularized, and the quality of health services and the level of protection will be continuously improved. (Lines 49-52)

(2)Lines 244-246

Original text：However, through the super-efficiency SBM measurement, it is found that the comprehensive efficiency score of the samples is 0.975, indicating that the allocation efficiency of medical and health resources in Hainan Province is not well, which needs to be further improved. 

The modified：However, through the super-efficiency SBM measurement, it is found that the comprehensive efficiency score of the samples is 0.975, which indicates that the efficiency of healthcare resource allocation in Hainan Province is not high and needs to be further improved.

(3)Line 248

Original text：The main reasons for the decline may have been these reasons.

The modified：The main reasons for the decline may be these:

(4)Lines 278-283

Original text：On the one hand, Hainan Province is not able to improve its medical and health output through scientific and technological progress and innovation. On the other hand, the current investment of the Hainan Provincial government in the medical and health field focuses too much on the hardware elements and neglects to some extent the improvement of medical and health technology, medical service quality, and health personnel quality.

The modified：The reason for this is that, on the one hand, Hainan Province is unable to improve healthcare output through scientific and technological progress and innovation. On the other hand, the current investment in healthcare by the government of Hainan Province is too focused on hardware elements, and to a certain extent, it has neglected the improvement of healthcare technology, the quality of healthcare services, and the quality of health personnel. 

(5)Lines 341-346

Original text：The Hainan provincial government should further implement and optimize policies, accelerate the process of medical reform, improve detailed medical and health resource planning, and rationally allocate medical and health hardware and software resources according to population, geography, demand, and other factors in order to improve efficiency.

The modified：The Hainan government should further implement and optimize its policies, accelerate the process of healthcare reform, improve detailed healthcare resource planning, rationally allocate healthcare hardware and software resources based on demographic, geographic, and demand factors, and promote the expansion and balanced allocation of high-quality healthcare resources to improve efficiency.

Thank you very much for your attention and time. Looking forward to your reply.

Yours sincerely,

Yanhua GONG

---

## [Decision Letter · Decision Letter 2]

18 Oct 2023

PONE-D-23-10563R2Study on the allocation efficiency of medical and health resources in Hainan Province: Based on the super-efficiency SBM — Malmquist modelPLOS ONE

Dear Dr. Gong,

Thank you for submitting your manuscript to PLOS ONE. After careful consideration, we feel that it has merit but does not fully meet PLOS ONE’s publication criteria as it currently stands. Therefore, we invite you to submit a revised version of the manuscript that addresses the points raised during the review process. Thnak you for the effort of updating the manuscript to meet Rewievers' requirements. Before I accept the paper to be published I suggest strongly to provide in the text reason for selecting research data span, discuss briefly the limitation on the methodology used and delete all the suggestions if they could not be directly derived from the Super-DEA models estimation result (or explain in the text that they do not come from DEA model). Specific details can be found in the Reviewers' comments below.

We look forward to receiving your revised manuscript.

Kind regards,

Agata Sielska, Ph.D.

Academic Editor

PLOS ONE

Journal Requirements:

Reviewers' comments:

Reviewer's Responses to Questions

**Comments to the Author**

1. If the authors have adequately addressed your comments raised in a previous round of review and you feel that this manuscript is now acceptable for publication, you may indicate that here to bypass the “Comments to the Author” section, enter your conflict of interest statement in the “Confidential to Editor” section, and submit your "Accept" recommendation.

Reviewer #2: All comments have been addressed

Reviewer #3: All comments have been addressed

2. Is the manuscript technically sound, and do the data support the conclusions?

Reviewer #2: Yes

Reviewer #3: Yes

3. Has the statistical analysis been performed appropriately and rigorously? 

Reviewer #2: Yes

Reviewer #3: Yes

4. Have the authors made all data underlying the findings in their manuscript fully available?

Reviewer #2: Yes

Reviewer #3: Yes

5. Is the manuscript presented in an intelligible fashion and written in standard English?

Reviewer #2: Yes

Reviewer #3: Yes

6. Review Comments to the Author

Reviewer #2: The authors respond honestly to reviewers' comments. Analysis of the functioning of health systems is a particularly important topic for modern societies. The experiences of other countries - even with completely different regulations - are an important source for reflection on work aimed at improving their own health care. Therefore, it seems to me that it is worth disseminating research results, their discussion and recommendations for decision-makers. Despite this, I still feel that the contribution to the analysis, i.e. the impact of the port and free trade, is very specific.

Reviewer #3: The authors answered my main concerns, and I suggest to accept this manuscript if the following comment could be responded. (2)I strongly suggest the authors to reveal the reason to select research data span beginning form 2016. If longer data span is available, it should be incorparated into the work. If no available data is there, pls tell the readers in the manuscript. (2) the limitation on the methodology used in this paper should be bfiefly discussed, e.g. the spatial distribution of allocative efficeincy? (3) the present policy suggestions are very long and of low correlation with the findings of this manuscript. Without the authors' calculation, major suggestions could also be "CORRECT". I suggest the authors to delete all the suggestions if they could not be directly derived from the Super-DEA models estimation result.

7. PLOS authors have the option to publish the peer review history of their article (what does this mean?). If published, this will include your full peer review and any attached files.

Reviewer #2: No

Reviewer #3: No

---

## [Author Response · Author response to Decision Letter 2]

30 Oct 2023

Dear Editor and Reviewers:

Thank you very much for your comments and professional advice on the manuscript (Title: Study on the allocation efficiency of medical and health resources in Hainan Province: Based on the super-efficiency SBM — Malmquist model. Number: PONE-D-23-10563R3). These opinions are helpful to improve the academic and rigorous nature of the manuscript. In the previous review, we carefully revised the manuscript and answered all the questions raised by the reviewers. We would like to thank all reviewers for their professional advice and recognition of our revised manuscript. Similarly, this time we are also holding a serious and steadfast learning attitude. According to your suggestions, we have carefully revised the manuscript. We hope the manuscript can be better. We have not changed the overall framework of the manuscript, but the "suggestions" section has been modified considerably. In a marked revision manuscript, we have highlighted your proposed revision in yellow. Also, in this reply, we have highlighted some of the content in gray so that you can look more clear. Specific modification details are as follows:

Reviewer #3:

Thank you very much for your recognition of the manuscript and your professional suggestions. The following are our amendments and replies.

1.I strongly suggest the authors to reveal the reason to select research data span beginning form 2016. If longer data span is available, it should be incorparated into the work. If no available data is there, pls tell the readers in the manuscript.

The author's answer：Thank you very much for your professional advice. We add this content to the "Source of data" section. Based on the policy background, data availability, and trend changes, this paper selects the time span from 2016 to 2020 to study the allocation efficiency of medical and health resources in Hainan Province. However, the study on the allocation efficiency of medical and health resources in Hainan Province is a continuous study, and it is necessary to constantly supplement the complete relevant data to improve the allocation efficiency to the greatest extent with the development of time. Therefore, we will continue to pay attention to its development. (Lines 130-148) Specific additions are as follows:

Added content: This paper combines policy background, data availability, and trend changes to choose the time span of 2016–2020 to study the healthcare resource allocation efficiency in Hainan Province. In terms of policy background, since 2016, the Chinese government has proposed a series of policy measures to promote the efficiency of healthcare resource allocation, and the implementation of these policies has had a positive impact on the efficiency of healthcare resource allocation in Hainan Province. In addition, in 2018, under the policy of the Free Trade Port, Hainan Province is fully committed to deepening reform. Therefore, the choice of 2016 as the starting year has a certain policy representativeness, and the choice of 2020 as the cut-off year is because when this study was done, the data could only be obtained up to 2020. In terms of data availability, after 2016, the data related to the allocation of healthcare resources in Hainan Province were gradually improved, which can reflect the actual situation of healthcare resource allocation in Hainan Province in a more comprehensive way, and there were more missing data before 2016. Therefore, we chose the time span of 2016–2020 to ensure the accuracy and reliability of the data as much as possible. In terms of trend change, with the passage of time, the trend of healthcare resource allocation efficiency in Hainan Province will also change, especially in the context of the free trade port. By studying the time span of 2016–2020, the changing trend of healthcare resource allocation efficiency in Hainan Province before and after the implementation of the Free Trade Port can be more clearly grasped, thus providing a reference for policy formulation in Hainan Province.

2.The limitation on the methodology used in this paper should be bfiefly discussed, e.g. the spatial distribution of allocative efficeincy?

The author's answer：Thank you very much for your professional advice. This led us to think more deeply. We have supplemented the limitations of super-efficiency SBM in the "Super-efficiency SBM" section. (Lines 194-199) The limitations of the Malmquist index are supplemented in the "Malmquist index" section. (Lines 215-219) The details are as follows:

Added content: 

However, the super-efficiency SBM model also has its limitations. First, this assessment method ignores some non-economic factors, such as environmental efficiency and social efficiency. Therefore, it does not fully assess the comprehensive efficiency. Secondly, it has high requirements for input and output data, but there are also data that are more difficult to obtain in practice, which may affect the assessment results. Third, it cannot reflect the spatial distribution of efficiency. (Lines 194-199) 

The Malmquist index assesses technical efficiency change by calculating the product of technical progress and efficiency change. However, the biggest limitation of the method is that it ignores the source of technical progress. In other words, it is not known whether technical progress is driven by internal innovation or external introduction. This problem may lead to errors in the assessment results. (Lines 215-219) 

3.The present policy suggestions are very long and of low correlation with the findings of this manuscript. Without the authors' calculation, major suggestions could also be "CORRECT". I suggest the authors to delete all the suggestions if they could not be directly derived from the Super-DEA models estimation result.

The author's answer：Thank you very much for your professional advice. We have made changes to the "suggestions" section. After careful consideration, we did not delete all the recommendations but kept the concise ones in light of our findings. According to our conclusions, there are obvious regional differences in the efficiency of healthcare resource allocation in Hainan Province. In addition, efficiency is affected by the dual factors of scale efficiency and technological progress, of which technological progress is the key factor. Therefore, we believe that the allocation efficiency can be improved by promoting the expansion and balanced allocation of high-quality medical resources, expanding the total number of personnel, improving the quality of personnel, and upgrading the level of technology. (Lines 365-376) Specific modifications are set out below:

Original text：Improving the efficiency of the allocation of medical and health resources is an important guarantee for promoting the development of the health industry of the Hainan Free Trade Port and an inevitable requirement for the high-quality construction of the Hainan Free Trade Port. In order to improve the efficiency of medical and health resources in Hainan Free Trade Port, the following suggestions are put forward. 

First, Hainan should continue to take institutional integration and innovation as the "starting point" to improve the ability of modern governance of health services. The Hainan government should further implement and optimize its policies, accelerate the process of healthcare reform, improve detailed healthcare resource planning, rationally allocate healthcare hardware and software resources based on demographic, geographic, and demand factors, and promote the expansion and balanced allocation of high-quality healthcare resources to improve efficiency. Accelerate the development of key health projects in Hainan Province. 

Second, Hainan should balance the layout and optimize the allocation of health resources, and speed up the process of promoting graded diagnosis and treatment and "Internet Plus" medical treatment. Strengthen supervision and management, appropriately expand the scale of medical services in big cities, and for inefficient cities and counties, and improve the efficiency of service utilization by means of technology introduction, graded diagnosis and treatment, and "Internet +" medical treatment while ensuring adequate input. Accelerate the implementation of the relevant implementation opinions of Hainan Province's grid-tight medical and health service system, improve the medical consortium, hospital group, and other division of labor and cooperation models, and comprehensively improve diagnosis and treatment efficiency and resource utilization efficiency. 

Third, accelerate the introduction and training of health professionals, expand the total number, and improve the quality. On the one hand, the establishment of first-class medical colleges, especially nursing colleges. Independently cultivate and stabilize health professionals. On the other hand, Hainan Province can rely on favorable policies to improve the salary of health professionals, innovate the post and staffing system, focus on introducing and training high-level health professionals, master the core medical technology, improve the level of medical technology, and create high-end medical services in the world. In addition, the policy should be appropriately tilted to the grass-roots level and accelerate the construction of the grass-roots health personnel team. 

Fourth, strengthen international cooperation and exchanges to promote the sharing of medical and health resources. To maintain close contact with Beijing, Shanghai, Guangdong Province, and other places to strengthen technical exchanges and improve the level of medical technology and service efficiency. The introduction of domestic and foreign first-class medical institutions and research centers in Hainan to achieve co-construction, co-governance, and sharing to serve the construction of the Hainan Free Trade Port.

The modified：Improving the efficiency of the allocation of medical and health resources is an important guarantee for promoting the development of the health industry of the Hainan Free Trade Port and an inevitable requirement for the high-quality construction of the Hainan Free Trade Port. According to the conclusions of the paper, in order to improve the efficiency of medical and health resources in Hainan Free Trade Port, Hainan Province should rationalize the allocation of medical hardware and software resources based on demographic, geographic, and demand factors, accelerate the promotion of hierarchical diagnosis and treatment and "Internet+" medical care, and promote the expansion and balanced allocation of high-quality medical resources to improve efficiency. In addition, Hainan Province should accelerate the introduction and cultivation of health professionals, focusing on the improvement of medical and health technologies, the quality of medical and health services, and the quality of medical and health personnel.

Thank you very much for your attention and time. Looking forward to your reply.

Yours sincerely,

Yanhua GONG

---

## [Decision Letter · Decision Letter 3]

9 Nov 2023

Study on the allocation efficiency of medical and health resources in Hainan Province: Based on the super-efficiency SBM — Malmquist model

PONE-D-23-10563R3

Dear Dr. Gong,

We’re pleased to inform you that your manuscript has been judged scientifically suitable for publication and will be formally accepted for publication once it meets all outstanding technical requirements.

Regarding the change of finance disclosures, I believe I cannot do this on my own. Please, contact the Journal's Office directly, they will be able to help you with that. I will also let them know about this issue.

Kind regards,

Agata Sielska, Ph.D.

Academic Editor

PLOS ONE

Additional Editor Comments (optional):

Reviewers' comments:

Reviewer's Responses to Questions

**Comments to the Author**

1. If the authors have adequately addressed your comments raised in a previous round of review and you feel that this manuscript is now acceptable for publication, you may indicate that here to bypass the “Comments to the Author” section, enter your conflict of interest statement in the “Confidential to Editor” section, and submit your "Accept" recommendation.

Reviewer #3: All comments have been addressed

2. Is the manuscript technically sound, and do the data support the conclusions?

Reviewer #3: Yes

3. Has the statistical analysis been performed appropriately and rigorously? 

Reviewer #3: Yes

4. Have the authors made all data underlying the findings in their manuscript fully available?

Reviewer #3: Yes

5. Is the manuscript presented in an intelligible fashion and written in standard English?

Reviewer #3: Yes

6. Review Comments to the Author

Reviewer #3: The present manuscript has fully improved, so I suggest to accept this present manuscript since my main concerns have been responded.

7. PLOS authors have the option to publish the peer review history of their article (what does this mean?). If published, this will include your full peer review and any attached files.

Reviewer #3: No

---

## [Editor Report · Acceptance letter]

14 Nov 2023

PONE-D-23-10563R3 

Study on the allocation efficiency of medical and health resources in Hainan Province: Based on the super-efficiency SBM — Malmquist model 

Dear Dr. Gong:

I'm pleased to inform you that your manuscript has been deemed suitable for publication in PLOS ONE. Congratulations! Your manuscript is now with our production department. 

Kind regards, 

on behalf of

Dr. Agata Sielska 

Academic Editor

PLOS ONE